

# Review of Gridded Climate Products and Their Use in Hydrological Analyses Reveals Overlaps, Gaps, and Need for More Objective Approach to Model Forcings

Kyle R. Mankin[1], Sushant Mehan[2], Timothy R. Green[1], David M. Barnard[1,3]

[1]Water Management & Systems Research Unit, USDA-Agricultural Research Service, Fort Collins, CO, USA
[2]Agricultural and Biosystems Engineering Department, South Dakota State University, Brookings, SD, USA
[3]Deparment of Ecosystem Science and Sustainability, Colorado State University, Fort Collins, CO, USA

*Correspondence to*: Kyle R. Mankin (kyle.mankin@usda.gov)

**Abstract.** Climate forcing data accuracy drives performance of hydrologic models and analyses, yet each investigator needs
to select from among the numerous gridded climate dataset options and justify their selection for use in a particular hydrologic
model or analysis. This study aims to provide a comprehensive compilation and overview of gridded datasets (precipitation,
air temperature, humidity, windspeed, solar radiation) and considerations for climate product selection criteria for hydrologic
modelling and analyses. All datasets summarized here span at least the coterminous U.S. (CONUS), and many are continental
or global in extent. Gridded datasets built on ground-based observations (17), satellite imagery (20), and/or reanalysis products
(23) are compiled and described, with focus on the characteristics that hydrologic investigators may find useful in discerning
acceptable datasets (variables, coverage, resolution, accessibility, latency). Best-available-science recommendations for
dataset selection are based on a review of 28 recent studies (past 10 years) that compared performance of various gridded
climate datasets for hydrologic analyses. No single best source of gridded climate data exists, but several common themes
arose. Gridded daily temperature datasets improved when derived over regions with greater station density. Similarly, gridded
daily precipitation data were more accurate when derived over regions with higher-density station data, when used in spatially
less-complex terrain, and when corrected using ground-based data. In mountainous regions as well as humid regions,
reanalysis-based datasets generally performed better than ground-based when underlying data had low station density, but for
higher station densities, there was no difference. Ground-based precipitation datasets generally performed better than satellite-
or reanalysis-based datasets, though better precipitation and temperature datasets did not always translate into better
streamflow modelling. Hydrologic analyses would benefit from improved gridded datasets that retain dependencies among
climate variables and better represent small-scale spatial variability of climate variables in complex topography.

## 1 Introduction

Hydrologists are faced with a dizzying variety of options when selecting climate data for water resource analyses. Climate
drives hydrological processes, and accurate climate forcing data are essential for meaningful water resource modelling and





analyses. However, it is arguable that no single source of climate data is universally appropriate. Over recent decades, while ground-based observations from weather stations have decreased (Sun et al., 2018; Strangeways, 2006), gridded datasets built on ground-based observations, satellite imagery, and reanalysis products have increased. Many studies have intercompared the accuracy of particular subsets of these gridded climate datasets for various regions, settings, and time frames across the globe with various insights and conclusions. However, most studies focus on only a limited number of datasets, lack generalizable

recommendations, and do not consider the functional implications of dataset limitations on end-users' hydrologic analysis. This study aims to provide a comprehensive compilation, overview, and considerations for selection of gridded datasets with focus on selection for hydrologic modelling and analyses. Our focus is on datasets at the conterminous U.S. (CONUS) to global extents.

## 2 Gridded Dataset Sources

A well-maintained, long-term weather station, though not error-free (Gebremichael, 2010; Strangeways, 2006), provides direct, in-situ point measurements for a location. However, most hydrologic analyses address processes at scales greater than a point, for which weather station data may not be representative. Gridded datasets offer several advantages over point station data (Essou et al., 2016a): gridded datasets are relatively easy to use, have uniform spatial coverage, provide consistent coverage over time (avoids the problem of non-reporting stations), and rarely have missing data. Uniform grids with temporal

consistency allow simple averaging across a domain. However, gridded datasets often are not available in real-time (i.e., data latency), which might pose limitations for some hydrologic analyses (e.g., snowmelt and runoff forecasting, operational water resource decision making).

Gridded climate datasets can be categorized as ground-based (G), satellite-based (S), or reanalysis-based (R) according to the sources of data and methods used in their derivation. Many datasets integrate multiple data sources and methods in deriving

the dataset; in this article, the primary data source/method in integrated datasets is listed first followed by secondary method(s) (e.g., SR, RG, RSG). We focus on gridded datasets available for five climatological variables that are essential to hydrological analyses: precipitation (P), air temperature (T), atmospheric moisture (relative humidity [rh], dew-point T [Tdp], or vapor pressure [Vp]), windspeed (u), and solar radiation (Rs). Particular emphasis is on datasets that provide gridded P, a highly variable and critical driver in hydrological analyses. For more details, the reader is directed to an informative review of global

P datasets, including a discussion of these dataset sources and estimation procedures (Sun et al., 2018).

Although the grid resolution of each data product is clear, the support scale is generally vague. That is, the grid centroid is often treated as a point, which is then interpolated or regionalized to obtain areal averaged values at the scale of hydrologic model resolution (e.g., a hydrologic response unit or HRU). However, if the gridded data represent grid-scale (e.g., 4 km by 4 km) areal averages, this should be considered during interpolation to the HRU scale. Scaling within and across grid cells has





been explored for gridded soil moisture (Hoehn et al., 2017), but remains an issue for gridded climate products. In this study, we mention this as a precaution but do not offer scaling solutions.

## 2.1 Ground-based (G)

Ground-based gridded datasets (**Table 1**) are derived directly from observational data, typically from weather station networks. Various methods are used to interpolate data between stations and may account for orographic effects, lake effects, and other

mesoscale meteorologic phenomena. These datasets benefit from direct application of data with relatively well-defined biases and uncertainty inherited from the instrumentation characteristics and errors. For example, P data collection has well-known errors at the station level from sources such as wind, evaporation, wetting, splashing, site location, instrument error, spatiotemporal variation in drop-size distribution, and frozen versus liquid P (Sun et al., 2018). Interpolating these data to a grid adds additional uncertainty to the extent that station density inadequately captures spatial variability of the climatic

variable across the domain. Minimum recommended station densities vary by physiographic unit (mountains, plains, etc.) from 1 to 4 stations per 1000 km$^2$ (WMO, 2008). Essou et al. (2017) noted that most of the 316 watersheds in their comprehensive Canadian study had less than 1 station per 1000 km$^2$, indicating a wider global concern. Increased station density generally improves gridded dataset quality, but it may be impractical to adequately cover regions with complex topography, localized convective storms, heat islands, blowing snow, or other micrometeorological heterogeneity. For example, snow gauge

undercatch due to high windspeeds is an especially pronounced phenomenon that challenges accurate characterization of water storage in snow dominated basins (Fassnacht, 2004; Panahi & Behrangi, 2020). Station density and coverage also change over time as old stations are deprecated or new stations added, complicating interpolation schemes and often disproportionately diminishing coverage in remote areas. Sun et al. (2018) noted that the number of global stations in the GPCC v7 dataset has changed from 10,900 stations in 1901, to a maximum of 49,470 in 1970, decreasing to 30,000 in 2005, and only 10,000 in

2012. This recent decline in station data not only impacts G datasets but also S and R datasets that rely on station data in their dataset development. Uncertainty associated with these temporal changes in sampling density are further complicated by nonstationarity of climate and accelerated climate change in recent decades.

**Table 1. Summary of ground-based (G) gridded datasets.**

| Dataset Name | Data Source | Variables | Spatial Resolution | Temporal Resolution | Spatial Coverage | Temporal Coverage | Latency | Data Format | Reference [Data Availability] |
|---|---|---|---|---|---|---|---|---|---|
| BEST | GR | T | 0.25°, 1° | monthly | Land Global | 1753-NP 1850-NP | months | NetCDF | Rohde & Hausfather (2020) [https://berkeleyearth.org/data/] |
| CPC | G | P | 0.25° | 24 h | Land | 1948-NP | 1 d | NetCDF | Chen et al. (2008), Xie et al. (2007) [https://ftp.cpc.ncep.noaa.gov/precip/CPC_UNI_PRCP/GAUGE_CONUS/] |
| CPC-Unified | G | P | 0.5° | 24 h | Land | 1979-NP | 1 d | NetCDF | Chen et al. (2008) [https://ftp.cpc.ncep.noaa.gov/precip/CPC_UNI_PRCP/GAUGE_GLB/RT] |





| | | | | | | | | | | |
|---|---|---|---|---|---|---|---|---|---|---|
| CRU-TS v4.6 | G | P, T | 0.5° | monthly | Land | 1901-2021 | irreg. | NetCDF | Harris et al. (2020) [https://data.ceda.ac.uk/badc/cru/data/cru_ts/cru_ts_4.06/data] |
| Daymet | G | P, T, Vp, Rs | 1 km | 24 h | CONUS | 1980-NP | CY | NetCDF | Thornton et al. (2021) [https://thredds.daac.ornl.gov/thredds/catalog/ornldaac/2129/catalog.html] |
| EMDNA | GR | P, T | 11 km | 24 h | N Amer | 1979-2018 | — | NetCDF | Tang et al. (2021) [https://gwfnet.net/Metadata/Record/T-2020-11-25-i1Fwxi32sBMU2GDhUZ6gAJEg] |
| GLDAS | GS | P | 0.125° | 3 h | Global | 2000-NP | 2 mo | NetCDF | Rodell et al. (2004) [https://hydro1.gesdisc.eosdis.nasa.gov/opendap/GLDAS/] |
| GPCC v7 | G | P | 0.25°, 0.5°, 1.0°, 2.5° | monthly | Land | 1891-2020 | — | NetCDF | Schneider et al. (2017, 2016) [https://opendata.dwd.de/climate_environment/GPCC/html/fulldata-monthly_v2022_doi_download.html] |
| GPCC-FDD | G | P | 1.0° | 24 h | Land | 1982-2020 | — | NetCDF | Schamm et al. (2014) [https://opendata.dwd.de/climate_environment/GPCC/html/fulldata-daily_v2022_doi_download.html] |
| gridMet | G | P, T, rh, u, Rs | 4 km | 24-h | CONUS | 1979-NP | 60 d | NetCDF | Abatzoglou (2013) [https://www.northwestknowledge.net/metdata/data/] |
| Livneh | G | P, T, u | 0.0625° | 24 h | CONUS | 1915-2011 | — | NetCDF | Livneh et al. (2013) [https://psl.noaa.gov/thredds/catalog/Datasets/livneh/metvars/catalog.html] |
| nClimGrid | G | P, T | 48.3 km | 24-h | CONUS | 1951-NP | 1 d | NetCDF | Durre et al. (2022) [https://www.ncei.noaa.gov/data/nclimgrid-daily/archive/] |
| NLDAS-2 | GR | P, T, rh, u, Rs | 0.125° | 1 h | N Amer | 1979-NP | 4 d | GRIB, NetCDF | Xia et al. (2012a, b) [https://hydro1.gesdisc.eosdis.nasa.gov/data/NLDAS/] |
| PRISM | G | P, T, rh | 4 km | 24 h | CONUS | 1895-NP | 1 yr | ASCII, NetCDF, GeoTIFF | Daly et al. (2008) [https://prism.oregonstate.edu/explorer/] |
| Santa-Clara | G | P, T | 0.125° | 24 h | CONUS | 1949-2010 | — | ASCII, NetCDF | Maurer et al. (2002) [https://www.engr.scu.edu/~emaurer/gridded_obs/index_gridded_obs.html] |
| TopoWx | GR | T | 0.8 km | 24 hr | CONUS | 1948-2017 | — | NetCDF | Oyler et al. (2015) [https://www.scrim.psu.edu/resources/topowx/] |
| UDEL | G | P, T | 0.5° | monthly | Land | 1900-2014 | — | NetCDF | Matsuura et al. (2023) [https://psl.noaa.gov/data/gridded/data.UDel_AirT_Precip.html] |

**Data Source**: G=ground-based observations (with interpolation), S=satellite, R=reanalysis. **Variables**: P=precipitation, T=air temperature,

rh=relative humidity, u=windspeed, Rs=solar radiation, Vp=vapor pressure. **Spatial Resolution**: 1.0° latitude=111 km, 1.0° longitude=111





km at 0° latitude and 85 km at 40° latitude. **Spatial Coverage**: Land=Global land surfaces only (not ocean surfaces), CONUS=contiguous U.S. **Temporal Coverage**: NP=near present. **Latency**: CY=Available each calendar year, ——=Static dataset. **Data Format**: NetCDF=Network Common Data Form, ASCII=American Standard Code for Information Interchange, GRIB=Gridded Binary, GeoTIFF=Georeferenced Tagged Image File Format.

## 2.2 Satellite-based (S)

Satellite-based gridded datasets (**Table 2**) are derived from various sensors onboard geostationary satellites (visible/infrared [IR] sensors) with rapid sampling frequency (30 minutes or less) and low-Earth orbit satellites (visible/IR, passive microwave [MW], and active MW) with lower temporal sampling frequency (Sun et al., 2018). Compared to G datasets, S datasets provide spatially homogenous coverage (the entire area within the coverage field has similar data density) and temporally continuous records but are limited in temporal coverage to the satellite era, with the first Television and IR Observation Satellite (TIROS) launched in 1960. Visible/IR methods detect cloud-top surface conditions and correlate colder/brighter cloud tops to greater convection and more P. Passive MW methods detect precipitation-sized particles, which provides a more-direct measure of P. Active MW methods allow measurement of the instantaneous three-dimensional structure of rainfall. Methods have been developed to merge these datasets to capitalize on the higher accuracy of MW methods and greater temporal frequency of visible/IR methods and increase overall product accuracy (Sun et al., 2018).

A review by Maggioni et al. (2016) described satellite instruments and compared many of the algorithms used in current satellite P datasets. Satellite-based products have larger overestimation bias in the warm season and lower positive bias in the cold season. Satellite datasets have high probability of capturing warm-season convective events; as a result, in the central U.S., for example, S datasets have better agreement with ground-radar products than rain-gauge stations, which can miss localized convective storms. Satellite-based products tend to underestimate intense rainfall during extreme hurricane events; S also tends to underestimate light P at high elevations and overestimate P at low elevations in regions of complex topography in northwestern Mexico and the Appalachian Mountains, all of which may be attributed to IR sensors' lack of discrimination between raining and non-raining clouds.

**Table 2. Summary of satellite-based (S) gridded datasets.**

| Dataset Name | Data Source | Variables | Spatial Resolution | Temporal Resolution | Spatial Coverage | Temporal Coverage | Latency | Data Format | Reference [Data Availability] |
|---|---|---|---|---|---|---|---|---|---|
| CHIRP v2 | SR | P | 0.05° | 24 h | Land, <50° | 1981-NP | 2 d | GeoTIFF | Funk et al. (2015) [https://data.chc.ucsb.edu/products/CHIRP/] |
| CHIRPS v2 | SRG | P | 0.05° | 24 h | Land, <50° | 1981-NP | 1 mo | GeoTIFF | Funk et al. (2015) [https://data.chc.ucsb.edu/products/CHIRPS-2.0/] |
| CMORPH v1 | S | P | 0.07°, 0.25° | 0.5 h, 24 h | <60° | 1998-NP | 5-6 mo | NetCDF | Joyce et al. (2004), Xie et al. (2017) [https://www.ncei.noaa.gov/data/cmorph-high-resolution-global-precipitation-estimates/; |



| | | | | | | | | | |
|---|---|---|---|---|---|---|---|---|---|
| CMORPH-BLD v1 | SG | P | 0.25° | 24 h | <60° | 2003-NP | 1 mo | GRIB, NetCDF | https://noaa-cdr-precip-cmorph-pds.s3.amazonaws.com/index.html] Sun et al. (2016) [https://ftp.cpc.ncep.noaa.gov/precip/CMORPH_V1.0/BLD/] |
| CMORPH-CRT v1 | SG | P | 0.07°, 0.25° | 0.5 h, 24 h | <60° | 1998-2015 | — | GRIB, NetCDF | Joyce et al. (2004), Xie et al. (2017) [https://ftp.cpc.ncep.noaa.gov/precip/CMORPH_V1.0/CRT/] |
| GPCPDAY/MON | SG | P | 0.5° | 24 h | Global | 2000-2021 | — | NetCDF | Huffman et al. (2023) [https://measures.gesdisc.eosdis.nasa.gov/data/GPCP/] |
| GPCP-1DD v1.2 | SG | P | 1.0° | 24 h | Global | 1996-2015 | — | NetCDF | Huffman et al. (2001) [https://rda.ucar.edu/datasets/ds728.3/dataaccess/] |
| GPM | SG | P | 0.1° | 0.5 h | <60° | 2014-NP | 24 h | HDF5, NetCDF | Hou et al. (2014) [https://gpm1.gesdisc.eosdis.nasa.gov/data/] |
| GSMaP v5/6 | S | P | 0.1° | 1 h | <60° | 2000-NP | 30 min | ASCII, GeoTIFF | Ushio et al. (2009), Kubota et al. (2020) [https://sharaku.eorc.jaxa.jp/GSMaP/] |
| IMERG-Early v6 | S | P | 0.1° | 0.5 h | Global | 2000-NP | 4 h | HDF5, NetCDF | Tan et al. (2019), Huffman et al. (2020a,b) [https://gpm1.gesdisc.eosdis.nasa.gov/data/GPM_L3/GPM_3IMERGDE.06/] |
| IMERG-Late v6 | S | P | 0.1° | 0.5 h | Global | 2000-NP | 14 h | HDF5, NetCDF | Tan et al. (2019), Huffman et al. (2020a,b) [https://gpm1.gesdisc.eosdis.nasa.gov/data/GPM_L3/GPM_3IMERGDL.06/] |
| IMERG-Final v6 | SG | P | 0.1° | 0.5 h | Global | 2000-NP | 3.5 mo | HDF5, NetCDF | Tan et al. (2019), Huffman et al. (2020a,b) [https://gpm1.gesdisc.eosdis.nasa.gov/data/GPM_L3/GPM_3IMERGDF.06/] |
| MSWEP v2.2 | SRG | P | 0.1° | 3 h | Global | 1979-NP | 3 h | NetCDF | Beck et al. (2017a, 2019) [https://www.gloh2o.org/mswep/] |
| NSRDB | SG | P, T, rh, u, Rs | 4 km | 1 h | CONUS | 1998-2021 | — | HDF5 | Sengupta et al. (2018), Buster et al. (2022) [https://nsrdb.nrel.gov/data-sets/how-to-access-data] |
| PERSIANN | SR | P | 0.25° | 1 h | <60° | 2000-NP | 1 h | NetCDF | Sorooshian et al. (2000) [https://persiann.eng.uci.edu/CHRSdata/PERSIANN/] |
| PERSIANN-CCS | S | P | 0.04° | 1 h | <60° | 2003-NP | 1-2 d | NetCDF | Hong et al. (2004) [https://persiann.eng.uci.edu/CHRSdata/PERSIANN-CCS/] |
| PERSIANN-CDR | SG | P | 0.25° | 24 h | <60° | 1983-NP | 1 mo | NetCDF | Ashouri et al. (2015) [https://www.ncei.noaa.gov/data/precipitation-persiann/access/2023/] |





| SM2RAIN-ASCAT | S | P | 0.1° | 24 h | Land | 2007-2021 | — | NetCDF | Brocca et al. (2014) [https://zenodo.org/records/7950103] |
| TMPA-3B42 v7 | SG | P | 0.25° | 3 h | <60° | 2000-2019 | — | NetCDF | Huffman et al. (2007), Gebremichael et al. (2010) [https://disc2.gesdisc.eosdis.nasa.gov/opendap/TRMM_L3/TRMM_3B42_Daily.7/] |
| TMPA-3B42RT v7 | S | P | 0.25° | 3 h | <60° | 1998-2019 | — | NetCDF | Huffman et al. (2007), Gebremichael et al. (2010) [https://disc2.gesdisc.eosdis.nasa.gov/opendap/TRMM_RT/TRMM_3B42RT.7/] |

**Data Source**: G=ground-based observations (with interpolation), S=satellite, R=reanalysis. **Variables**: P=precipitation, T=air temperature, rh=relative humidity, u=windspeed, Rs=solar radiation. **Spatial Resolution**: 1.0° latitude=111 km, 1.0° longitude=111 km at 0° latitude and 85 km at 40° latitude. **Spatial Coverage**: Land=Global land surfaces only (not ocean surfaces). **Temporal Coverage**: NP=near present. **Latency**: —=Static dataset. **Data Format**: NetCDF=Network Common Data Form, HDF5=Hierarchical Data Format 5, ASCII=American Standard Code for Information Interchange, GRIB=Gridded Binary, GeoTIFF=Georeferenced Tagged Image File Format.

## 2.3 Reanalysis-based (R)

Reanalysis-based gridded datasets (**Table 3**) are synthesized from process-based climate models, often together with G and/or S observational data, with the goal of generating gridded datasets with spatially homogenous data density that are temporally continuous. A precipitation forecast is generated from complex interactions of *a priori* predictions from a physically based, dynamical process model (that can often account for orographic effects in topographically complex regions) and ingested
observational data. Reanalysis systems use various models, observational datasets, and assimilation methods, can generate many climate variables with inter-dependent variable consistency, and provide near-real-time datasets with latency periods from hours to months. Accuracy of R methods may be limited by the changing availability of observational data and biases in observations and models.

Reanalysis datasets have been found to better capture winter P resulting from large-scale systems than summer P with greater
influence of localized convective storms (Massmann, 2020; Beck et al., 2019). Similarly, Beck et al. (2017b) confirmed the conclusions of several other studies (Barrett et al., 1994; Xie and Arkin, 1997; Adler et al., 2001; Ebert et al., 2007; Massari et al., 2017) that demonstrate reanalysis underperformed MW- and IR-based datasets in the tropics and outperformed them in colder regions (> 40° latitude). Reanalysis demonstrated reduced bias compared to S datasets, with greater ranges of bias among all datasets in areas with complex topography (Rockies, Andes, and Hindu Kush) and arid regions (Sahara and the
Arabian and Gobi deserts) (Beck et al., 2017b).

**Table 3. Summary of reanalysis-based (R) gridded datasets.**

| Dataset Name | Data Source | Variables | Spatial Resolution | Temporal Resolution | Spatial Coverage | Temporal Coverage | Latency | Data Format | Reference [Data Availability] |
| --- | --- | --- | --- | --- | --- | --- | --- | --- | --- |





| 20CR | R | P, T, rh, u, Rs | 1.0° | 3 h, 24 h | Global | 1836-2015 | — | NetCDF | Compo et al. (2011) [https://psl.noaa.gov/thredds/catalog/Datasets/20thC_ReanV3/miscSI/catalog.html] |
| CERA-20C | R | P, T, rh, u | 0.125° | 24 h | Global | 1901-2010 | — | NetCDF | Laloyaux et al. (2018) [https://apps.ecmwf.int/archive-catalogue/?class=ep] |
| ERA-20C | R | P | 125 km | 3 h | Global | 1900-2010 | — | GRIB | Poli et al. (2016) [https://thredds.rda.ucar.edu/thredds/catalog/aggregations/g/ds626.0/5/catalog.html] |
| ERA5 | R | P, T, rh, u, Rs | 0.25° | 1 h | Global | 1979-NP | 6 d | GRIB, NetCDF | Hersbach et al. (2018, 2020) [https://thredds.rda.ucar.edu/thredds/catalog/files/g/ds633.0/catalog.html] |
| ERA-Interim | RS | P, T, rh, u, Rs | 0.75° | 3 h | Global | 1979-NP | months | GRIB | Dee et al. (2011) [https://thredds.rda.ucar.edu/thredds/catalog/catalog_ds627.0.html] |
| EWEMBI v1.1 | RG | P, T, rh, u, Rs | 0.5° | 24 h | Global | 1976-2013 | — | NetCDF | Warszawski et al. (2014) [https://data.isimip.org/10.5880/pik.2019.004] |
| GFD-HYDRO | RSG | P | 0.5° | 3 h | Global | 1979-NP | 5 d | NetCDF | Berg et al. (2018, 2021) [https://zenodo.org/records/3871707] |
| GRASP | R | P, T | 1.125° | 24 h | Global | 1961-2010 | — | ? | Iizumi et al. (2014) [Available upon request.] |
| GSMaP-RNL | RG | P | 0.1° | 24 h | <60° | 2001-2013 | — | NetCDF | Kubota et al. (2007), Iguchi et al. (2009) [https://thredds-x.ipsl.fr/thredds/catalog/FROGs/GSMAP-gauges-RNLv6.0/catalog.html; https://thredds-x.ipsl.fr/thredds/catalog/FROGs/GSMAP-nogauges-RNLv6.0/catalog.html] |
| GSMaP-std v6 | RG | P | 0.1° | 24 h | <60° | 2001-2013 | — | NetCDF, GeoTIFF | Ushio et al. (2019), Kubota et al. (2020) [https://sharaku.eorc.jaxa.jp/GSMaP/] |
| JRA-55 | R | P | 0.56° | 3 h | Global | 1958-NP | days | GRIB | Kobayashi et al. (2015), Harada et al. (2016) [https://thredds.rda.ucar.edu/thredds/catalog/catalog_ds628.0.html] |
| MERRA | R | P, T, rh, u; Rs | 0.67°x0.5°; 1.0°x1.25° | 1 h (6 h?); 3 h | Global | 1979-2016 | — | HDF | Rienecker et al. (2011) [https://disc.gsfc.nasa.gov/datasets?page=1&project=MERRA] |
| MERRA-2 | RSG | P, T, rh, u | 0.625°x0.5° | 1 h | Global | 1980-NP | 2 mo | NetCDF | Gelaro et al. (2017), Reichle et al. (2017) [https://disc.gsfc.nasa.gov/datasets?keywords=MERRA-2%20Products&page=1] |





| NASA-POWER | RS | Rs, P, T | 0.625°x0.5°, 1.0° | 24 h | Global | 1980-NP | 14 h - 3 mo | ASCII, CSV, NetCDF, GeoTIFF | Zhang et al. (2009) [https://power.larc.nasa.gov/data-access-viewer/] |
| NCEP-CFSR | RS | P, T, rh, u, Rs | 0.3°, 0.5°, 1.0°, 1.9°, 2.5° | 6 h | Global | 1979-2011 | — | GRIB | Saha et al. (2010), Decker et al. (2012) [https://thredds.rda.ucar.edu/thredds/catalog/files/g/ds093.0/catalog.html] |
| NCEP-CFSR v2 | RS | P, T, rh, u, Rs | 0.2°, 0.5°, 1.0°, 2.5° | 6 h | Global | 2011-NP | days | GRIB | Saha et al. (2014) [https://thredds.rda.ucar.edu/thredds/catalog/files/g/ds094.0/catalog.html] |
| NCEP-NARR | RG | P, T, rh, u, Rs | 32 km | 3 h | N Amer <50° | 1979-NP | months | GRIB | Mesinger et al. (2006) [https://thredds.rda.ucar.edu/thredds/catalog/files/g/ds608.0/catalog.html] |
| PGMFD v2.1 | RG | P, T, rh, u, Rs | 0.5° | 24 h | Global | 1901-2012 | — | NetCDF | Sheffield et al. (2006) [https://data.isimip.org/search/simulation_round/ISIMIP2a/product/InputData/climate_forcing/princeton/] |
| PGF v3 | RG | P, T | 0.25° | 3 h | Global | 1948-2012 | — | NetCDF | Sheffield et al. (2006) [https://hydrology.soton.ac.uk/data/pgf/] |
| S14FD | R | P, T | 0.5° | 24 h | Global | 1958-2013 | — | NetCDF | Iizumi et al. (2017) [https://search.diasjp.net/en/dataset/S14FD] |
| WFDEI | R | P, T, rh, u, Rs | 0.5° | 3 h | Global | 1979-2016 | — | NetCDF | Weedon et al. (2014) [https://thredds.rda.ucar.edu/thredds/catalog/files/g/ds314.2/catalog.html] |
| WFD-20C | R | P, T, rh, u, Rs | 0.5° | 6 h | Global | 1901-2016 | — | NetCDF | Weedon et al. (2011) [https://data.isimip.org/search/simulation_round/ISIMIP2a/product/InputData/climate_forcing/watch-wfdei/; https://www.data.gov.uk/dataset/a83eef6d-30d3-479d-90b3-40c09c26d42c/watch-forcing-data-wfd-20th-century-tair-air-temperature-1901-2001] |
| WRFCONUS404 | R | P, T, rh, u, Rs | 4 km | 1h | CONUS | 1980-2021 | — | NetCDF | Liu et al. (2017) Rasmussen et al. (2023) [https://www.sciencebase.gov/catalog/item/6372cd09d34ed907bf6c6ab1; https://app.globus.org/file-manager?origin_id=39161d64-419d-4cc4-853f-f6e737644eb4&origin_path=%2F] |

**Data Source**: G=ground-based observations (with interpolation), S=satellite, R=reanalysis. **Variables**: P=precipitation, T=air temperature, rh=relative humidity, u=windspeed, Rs=solar radiation. **Spatial Resolution**: 1.0° latitude=111 km, 1.0° longitude=111 km at 0° latitude and 85 km at 40° latitude. **Spatial Coverage**: Land=Global land surfaces only (not ocean surfaces). **Temporal Coverage**: NP=near present.





**Latency**: —=Static dataset. **Data Format**: NetCDF=Network Common Data Form, HDF5=Hierarchical Data Format 5, ASCII=American Standard Code for Information Interchange, GRIB=Gridded Binary, GeoTIFF=Georeferenced Tagged Image File Format, CSV=comma separated variable, ?=unknown.

## 2.4 Integrated Products

Inherent limitations of individual data sources (G, S, or R) can be reduced by merging other data sources to reduce errors. Some reanalysis datasets are used independently or merge multiple reanalysis products (denoted by R in **Table 3**). Reanalysis datasets commonly ingest ground-based observational data (RG), satellite data (RS), or both (RSG). Some S datasets also integrate G data (denoted by SG in **Table 2**), reanalysis data (SR), or both (SRG) to enhance accuracy and reduce bias. Several data sources, CHIRP, CMORPH, IMERG, PERSIANN, and TMPA, offer multiple products with increasing data source complexity, often with increased latency and different spatial and temporal resolutions.

## 3 Considerations for Use of Gridded Dataset for Hydrologic Analyses

Gridded datasets summarized in **Tables 1, 2, and 3** span 0.8 to 278 km spatial resolutions, 0.5 to 720 h (monthly) temporal resolutions, 0.02 (30 min) to 365 d latencies, CONUS to global spatial coverage, and 10 to 271 year periods of record, starting as early as 1753 (**Figure 1**). Differences have emerged in the representation of G, S, and R datasets across many of these categories. G datasets have the finest spatial resolutions (1 km) and longest periods of record (>240 years), and tend to have the longest latency (average for G=105 d, compared to S=29 d and R=37 d). A greater proportion of G datasets have less-extensive spatial coverage (CONUS to North American continental in this study). S datasets start no earlier than 1979 and no greater than 45-year period of record (through 2023). More R and S datasets have finer temporal resolution than G datasets, with average resolutions of 10 d for R and 9 d for S compared to 185 d for G.

No single best source of gridded climate data exists. Many characteristics of gridded datasets influence the best product. We highlight several of the most important considerations in differentiating among the many possible gridded datasets. Most of these characteristics are detailed for each gridded dataset in **Tables 1, 2, and 3**.





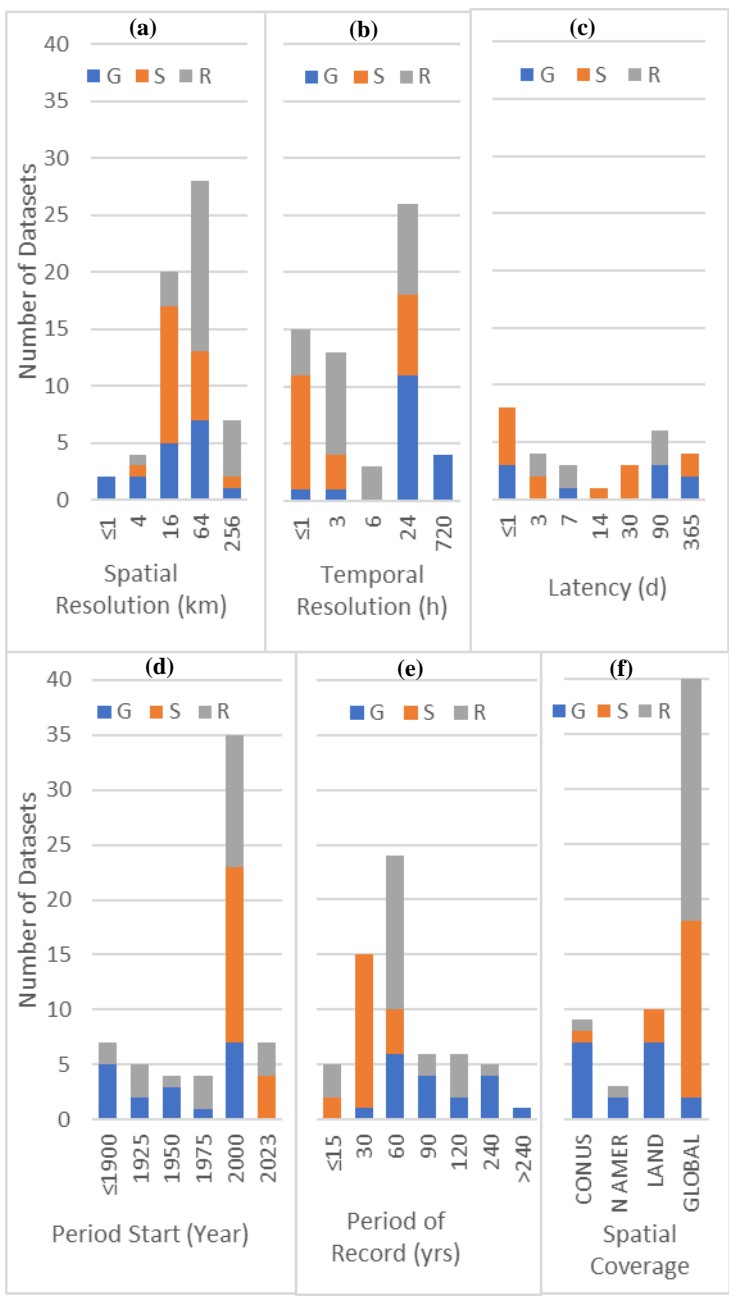

**Figure 1. Distribution of datasets among categories of (a) spatial resolution, (b) temporal resolution, (c) latency, (d) period start date, (e) period of record, and (f) spatial coverage for assessed ground-based (G), satellite-based (S), and reanalysis-based (R) gridded precipitation data sources. Numerical X axis labels are the upper limits of each categorical bin, exclusive of other bins.**





### 3.1 Variables and Interdependencies

Hydrological investigations typically begin with selecting datasets for each important climate variable. **Tables 1, 2, and 3**
summarize the variables included in each gridded dataset. Datasets that include all climate variables of interest may inherently represent appropriate interdependencies or cross-correlations among the variables. For example, periods of P are associated with cloud cover and decreased Rs and often higher humidity. Often these interdependencies are important to hydrologic analyses.

### 3.2 Coverage

Gridded datasets have a range of spatial and temporal extents. All datasets summarized in this article span at least the CONUS, and many are continental or global in extent. General guidelines by the World Meteorological Organization require a 30-year minimum period of record to reasonably represent climate variability. Non-stationarity of climate makes it even more important to consider whether longer periods representing climatic trends or periods more heavily weighted toward recent data are preferable for the given hydrologic study. Periods of record may be dictated by investigations focused on specific events
or periods, such as studies of the hydrologic effects of wildfire or other disturbance events or studies assessing hydrologic responses over specific periods.

### 3.3 Resolution

Spatial and temporal resolution of the dataset should be adequate to represent the variability of the climate variable given the representational scale of the hydrologic model. Some gridded datasets sacrifice representation of extremes, both wet and dry,
to better represent mean climatic conditions. Alternatively, increased temporal resolution often comes at the cost of reduced spatial resolution. Creation of spatially continuous and consistent gridded response surfaces can result in point data extremes being smoothed during interpolation. Methods that create ensembles of multiple gridded datasets often better represent mean conditions across a domain at the expense of representing the full range of possible conditions within the domain.

### 3.4 Format and Accessibility

Data format and accessibility dictate how easily and effectively a dataset can be accessed, processed, and analysed for a specific hydrologic application. Several common data formats are described in **Table A1**. The data format must be compatible with software and tools used in the hydrologic analysis. Formats such as NetCDF and HDF5 are widely used in climate research because they are consistent with various processing tools and can efficiently store large multidimensional datasets. Adequate metadata are essential for understanding the dataset, including its origin, methodology, and any processing it has undergone.
Investigators may consider the importance of datasets that can be compressed without significant data loss, are interoperable with the other datasets, and are freely available and easily accessible online. Some datasets have application programming interfaces (APIs) for automated data retrieval that can be useful.



### 3.5 Site and Event Characteristics

Preference may be given to datasets that reflect characteristic spatial and temporal dimensions of climatic processes in the
domain, such as cool-air drainage patterns, orographic or convective P events, lake effects, and effects of altitude. Priority may
be given to datasets that capture the most important aspects of climate variable magnitude and variability at appropriate scales,
including daily/seasonal/annual averages, extreme event (high or low) magnitudes, or event sequences (continuous dry days
[CDDs], continuous wet days [CWDs], etc.). For example, datasets with fine temporal resolution (~1 h) may be required to
capture hydrological functioning when P is dominated by high intensity but short duration convective events. In hydrologic
models, space-time scales are interdependent, and source data should be considered in watershed delineation.

### 3.6 Process and Model Sensitivity, and Latency

Hydrologic processes are differentially sensitive to climatic variables and characteristics. For example, a snowmelt runoff
modelling study may prioritize a dataset with accurate, fine-spatial-resolution T and accurate Rs whereas a small-basin study
of soil moisture or erosion dynamics may prioritize a fine-scale P dataset that maintains a full range of extreme events. A study
focused on evapotranspiration (ET) dynamics may prioritize a dataset that includes T, rh, u, and Rs and maintains appropriate
inter-variable dependencies. Flood simulation may prioritize fine-temporal-resolution P data at a resolution matching the
domain heterogeneity. Long-term water balance studies or large-scale river basin studies may prefer daily or monthly datasets
with coarse spatial resolution. Often the selected model formulation will constrain the required variables, their characteristics,
and the preferred data format.

Latency, or the time lag in dataset availability, also may be an important consideration. Some modelling applications may also
require real-time or near-real-time results. Gridded datasets may implement additional processing steps intended to increase
accuracy or resolution but that increase the latency before data become available for use.

### 3.7 Time Zone Considerations

When using climate and other hydrologic data from different sources, data time-period consistency is critical and too often
overlooked. Particularly for data in a "daily" format, users must be cognizant of the zonal time period for each dataset. Station
data varies on the reporting period for G data (e.g., daily periods beginning at midnight, 07:00 or 08:00 standard time or local
time [i.e., with spring and fall daylight savings time shifts]). Gridded datasets may provide data for a standard time period
(e.g., 24-hour period from 0:00 GMT) or adjusted for user-defined time zone. Hydrologic comparison datasets (e.g.,
streamflow) may be reported for 24 hours starting midnight standard or local time or for some other 24-hour period.
Mismatched datasets may lead to systematic analysis errors.



## 4 Review of Gridded Dataset Performance

Appropriate selection from among the many available gridded meteorological datasets requires an understanding of how these datasets impact hydrologic modelling. To assist with the selection process, we review and synthesize the recent (past 10 years) literature comparing gridded meteorological datasets, with specific consideration of their influence on hydrologic modelling
(**Table 4**). Studies were selected that (a) compared multiple gridded datasets, preferably including comparisons with different resolutions, scales, spatial contexts (topography, climate), goals, or hydrologic models; (b) compared the accuracy of those datasets to observed meteorological data; and (c) compared the performance of those datasets as forcing data for hydrologic model(s) or analyses. Most studies assessed and compared P datasets, some also assessed T datasets, and very few assessed rh, u, or Rs datasets. This relates, in equal measures, to the relative importance of P data in hydrologic analysis, the relative
complexity of representing P in gridded datasets, and the relative availability of P, T, and other data across G, S, and R datasets (**Tables 1, 2, 3**).

**Table 4. Summary of recent (10 years, 2014-2023) literature on gridded dataset comparisons.**

| Reference [Location] | Dataset Name | Data Source | Spatial Extent | Temporal Extent | Analysis Goals | Hydrologic Model | Hydrologic Outcomes |
|---|---|---|---|---|---|---|---|
| Ang et al. (2022) [SE Asia] | APHRODITE<br>NCEP-CFSR<br>TMPA-3B42 v7<br>IMERG-Final v6<br>ERA5<br>SA-OBS<br>CPC | G<br>R<br>SG<br>SG<br>R<br>G<br>G | 83,107 km², 100-1,700 m asl, 1,354 mm/y P | 1985-2011 | Compare P, T datasets to gauge stations and evaluate Q, ET performance | SWAT, daily, 0.25° grid | Good P (APHRODITE, ERA5, TMPA, IMERG), Good T (CPC, SA-OBS). TMPA and IMERG P with SA-OBS T provide reliable Q, ET. |
| Beck et al. (2017b) [Global] | CHIRP v2<br>CMORPH v1<br>ERA-Interim<br>GSMaP v5/6<br>GridSat v1<br>JRA-55<br>MSWEP-ng v1.2<br>MSWEP-ng v2<br>NCEP-CFSR<br>PERSIANN<br>PERSIANN-CCS<br>SM2RAIN-ASCAT<br>TMPA-3B42RT v7<br>CHIRPS v2<br>CMORPH-CRT v1<br>CPC-Unified<br>GPCP-IDD v1.2<br>MSWEP v1.2<br>MSWEP v2 | SR<br>S<br>R<br>S<br>S<br>R<br>SR<br>SR<br>R<br>S<br>S<br>S<br>S<br>SRG<br>SG<br>G<br>SG<br>SRG<br>SRG | 76,086 P stations; 9,035 basins (<50,000 km²) | P: 2000-2016, Q: 2000-2012 | Compare P datasets to daily gauge stations and evaluate daily Q performance (via 3-day NSE) | HBV, daily, conceptual | G best (CPC), but not transferable to low gauge density areas. Next best was SRG, with direct G correction (MSWEP). Among non-G corrected, SR were best (MSWEP) followed by R (ERA, MRA, NCEP) then SR (CHIRP). |





|  |  |  |  |  |  |  |  |
|---|---|---|---|---|---|---|---|
|  | PERSIANN-CDR v1r1 | SG |  |  |  |  |  |
|  | TMPA-3B42 v7 | SG |  |  |  |  |  |
|  | WFDEI-CRU | RG |  |  |  |  |  |
| Dembélé et al. (2020) [W Africa] | TAMSAT v3 | SG | 415,000 km², <400 m asl. | 2000-2012 | Calibration of daily Q, monthly Ea, Su, St | mHM, daily, 0.25° (28 km) discretization. | Best performing P datasets differed for: **Q** (TAMSAT, CHIRPS, PERSIANN-CDR), **temporal Su** (EWEMBI, WFDEI-GPCC, PGF), **spatial Su** (MSWEP, TAMSAT, ARC) **temporal Ea** (ARC, RFE, GSMaP), and **spatial Ea** (MSWEP, TAMSAT, MERRA-2) |
|  | CHIRPS v2 | SRG |  |  |  |  |  |
|  | ARC v2 | SG |  |  |  |  |  |
|  | RFE v2 | SG |  |  |  |  |  |
|  | MSWEP v2.2 | SRG |  |  |  |  |  |
|  | GSMaP-std v6 | RG |  |  |  |  |  |
|  | PERSIANN-CDR | SG |  |  |  |  |  |
|  | CMORPH-CRT v1 | SG |  |  |  |  |  |
|  | TMPA-3B42RT v7 | S |  |  |  |  |  |
|  | TMPA-3B42 v7 | SG |  |  |  |  |  |
|  | JRA-55 | R |  |  |  |  |  |
|  | EWEMBI v1.1 | RG |  |  |  |  |  |
|  | WFDEI-CRU | RG |  |  |  |  |  |
|  | WFDEI-GPCC | RG |  |  |  |  |  |
|  | MERRA-2 | RSG |  |  |  |  |  |
|  | PGF v3 | RG |  |  |  |  |  |
|  | ERA5 | R |  |  |  |  |  |
| Essou et al. (2016a) [CONUS] | MOPEX | G | 424 basins (66-10,325 km²), 5 climate regions | 1980-2003 | Comparison among observed climate data and simulated Q | HSAMI, daily, conceptual | Differences in P and T did not translate to differences in Q. |
|  | Santa-Clara | G |  |  |  |  |  |
|  | CPC | G |  |  |  |  |  |
|  | Daymet | G |  |  |  |  |  |
| Essou et al. (2016b) [CONUS] | Santa-Clara | G | 370 basins (104-10,325 km²), 5 climate regions | 1979-2003 | Comparison to observed climate data (Santa Clara) and Q (MOPEX) | HSAMI, daily, conceptual | Overall, global reanalyses were good proxies for observed P and T data. |
|  | ERA-Interim | R |  |  |  |  |  |
|  | NCEP-CFSR | R |  |  |  |  |  |
|  | MERRA | R |  |  |  |  |  |
|  | NCEP-NARR | RG |  |  |  |  |  |
|  | WFDEI-CRU | RG |  |  |  |  |  |
|  | WFDEI-GPCC | RG |  |  |  |  |  |
| Essou et al. (2017) [Canada] | ERA-Interim | R | 316 basins (440-127,635 km²), 3 climate regions | 1979-2010 | Compare P, T reanalyses to NRCan, and Q to CANOPEX. | HSAMI, daily, conceptual | Reanalysis performs better than gridded for low station density (1 per 1000 km²). |
|  | NCEP-CFSR | R |  |  |  |  |  |
|  | MERRA | R |  |  |  |  |  |
|  | NRCan | G |  |  |  |  |  |
| Gampe & Ludwig (2017) [Italy] | MESAN | DRG | 12,100 km², 0-3865 m asl, 500-1600 mm/y P | 1989-2008 | Comparison to observed climate data. | WaSiM, daily, 1 km resolution (not used in this study) | Recommend using an ensemble, excluding datasets with seasonal deviations (PERSIANN, ERA-Interim, ERA-20C). |
|  | APGD | G |  |  |  |  |  |
|  | E-OBS | G |  |  |  |  |  |
|  | PERSIANN-CDR | SG |  |  |  |  |  |
|  | MERRA-2 | RSG |  |  |  |  |  |
|  | ERA-Interim | R |  |  |  |  |  |
|  | GPCC-FDD | G |  |  |  |  |  |
|  | ERA-20C | R |  |  |  |  |  |





| Gupta & Tarboton (2016) [W US] | MERRA<br>RFE v2 | R<br>SG | 1,000,000 km² region | 2009-2010 | Compare downscaled climate data to SNOTEL. | UEB snowmelt (SWE), 3 h, 120 m climate downscale. | Good SWE simulation (NSE-0.67). Downscaling limitations noted. |
|---|---|---|---|---|---|---|---|
| Hafzi & Sorman (2022) [Turkey] | CPC v1<br>MSWEP v2.8<br>ERA5<br>CHIRPS v2<br>CHIRP v2<br>IMERG-Early v6<br>IMERG-Late v6<br>IMERG-Final v6<br>TMPA-3B42RT v7<br>TMPA-3B42 v7<br>PERSIANN-CDR<br>PERSIANN-CCS<br>PERSIANN | G<br>SRG<br>R<br>SRG<br>SR<br>S<br>S<br>SG<br>SG<br>S<br>SG<br>S<br>S | 10,250 km², 1130-3500 m asl | 2015-2019 | Evaluate climate data consistency and simulated Q | TUW, daily, conceptual | Most gridded P data were poor, but Q simulation quite accurate. Recommend calibrating Q model with same gridded data used to run simulation (not observed P data). |
| Henn et al. (2018) [W US] | H10<br>L15<br>PRISM-M<br>NLDAS-2<br>N15<br>Daymet | G<br>G<br>G<br>G<br>G<br>G | Western US (32-49°N, 105-125°W) | 1982-2006 | Intercompare spatial patterns, interannual variability, and multi-year trends in P | Limited comparison to Swe, Q. | Differences among datasets (especially high elevation, arid) may introduce substantial uncertainty. |
| Kouakou et al. (2023) [W, C Africa] | ARC v2<br>CHIRP v2<br>CHIRPS v2<br>PERSIANN-CDR<br>MSWEP v2.2<br>TAMSAT v3<br>ERA5<br>JRA-55 Adj<br>MERRA-2 P-TOT<br>MERRA-2 P-COR<br>WFDEI-CRU<br>WFDEI-GPCC<br>CPC v1<br>CRU-TS v4<br>GPCC v7 | SG<br>SR<br>SRG<br>SG<br>SRG<br>SG<br>R<br>RG<br>RSG<br>RSG<br>RG<br>RG<br>G<br>G<br>G | 68 basins (1,279-600,000 km²), 200-5,000 mm/y P | 1984-2005 | Evaluate P datasets, monthly Q simulation. | GR2M, monthly, lumped | Best P from G datasets. CHIRPS best for Q. |
| Laiti et al. (2018) [Italy] | E-OBS<br>MSWEP<br>MESAN<br>APGD<br>ADIGE | G<br>SRG<br>DRG<br>G<br>G | 12,100 km², 185-3,500 m asl | 1989-2008 | Assess hydrologic coherence of gridded data for daily Q. | HYPERstream + SCS-CN, daily, 5 km grid | The higher-res G datasets had best Q. |





| Massman (2020) [CONUS] | CERA-20C 20CR Livneh | R R G | 168 basins | 1900s-2010s | Assess century datasets for P, T (vs. Daymet), Q simulation | HBV, daily, conceptual | Quality decreases further back in history; T better than P. G better than R. |
|---|---|---|---|---|---|---|---|
| Mazzoleni et al. (2019) [Global] | CHIRP v2 CMORPH v1 PERSIANN PERSIANN-CCS SM2RAIN-ASCAT TMPA-3B42RT v7 CHIRPS v2 CMORPH-CRT v1 GPCP1DD v1.2 MSWEP v2.1 PERSIANN-CDR TMPA-3B42 v7 CPC Glob Unified GPCC GSMaP-RNL PFD WFDEI CRU WFDEI GPCC | SR S S S S S SRG SG SG SRG SG SG RG RG RG RG RG RG | 8 basins, 200-6,150,000 km², tropical to temperate climate zones | 2007-2013 | Compare P datasets for Q simulation, assess P density, model effects | HBV-96, daily?, conceptual, ~0.25° grid | No single best P dataset. Basin characteristics important. Q affected by basin scale, human footprint, climate. S poorest, most variable. SG best in Tropical, Temperate-arid climates. RG best in Temperate, Temperate-cold climates and densely gauged P basins. Subbasins had different best P dataset than outlet (distributed model better than lumped). |
| Mei et al. (2022) [Texas] | TMPA-3B42 NCEP-CFSR PRISM | SG R G | 535.76 km², 176-548 m asl | 1989-2009 | Compare P data to NOAA, modeled Q in urban basin. | SWAT, daily, subbasins ~21.4 km² (~4.6 km)², and ANN, daily, lumped | PRISM had best P, TMPA underestimated P, PRISM and TMPA outperformed CFR or gauge data for Q. SWAT and ANN had similar results for same P data. |
| Meng et al. (2014) [NE Tibetan Plateau] | TMPA-3B42 v6 | SG | 122,000 km², 4,000 m asl, 250-750 mm/y P | 1998-2008 | Compare P to NCC gauge and simulated Q | CREST, daily, distributed, 1 km² grid | TMPA daily P less than monthly P. TMPA unsatisfactory for daily Q, acceptable for monthly Q simulation. |
| Mourtzinis et al. (2017) [Midwest US] | Daymet PRISM NASA-POWER | G G RS | 45 stations in US Corn Belt. | 1980-2014 (12-35 yrs) | Compare ETo calculated with station and gridded data. | FAO-PM ETo. | Poor ETo related to poor rh, esp. for PRISM. |
| Muche et al. (2020) [Kansas] | Daymet PRISM NLDAS GLDAS | G G G G | 2,988 km², 252-428 m asl | 1983-2013 | Compare to GHNC. Calibrate monthly Q, simulate daily Q. | SWAT, daily, subbasins ~76.6 km² (~8.8 km)² | All monthly Q simulation similar (except GLDAS). |





| Pokorny et al. (2020) [Canada] | ANUSPLIN NCEP-NARR ERA-Interim WFDEI GFD-HYDRO | G RG RS RG RSG | 1,400,000 km² basin (7 subbasins), diverse climate regions | 1984-2010 | Compare P data aggregations. | -- | All gridded datasets showed spatial performance variations. Some aggregation reduces input uncertainty, but info lost as aggr. incr. |
|---|---|---|---|---|---|---|---|
| Radcliffe & Mukundan (2017) [Georgia] | PRISM NCEP-CFSR | G R | 44.7 km² | 2003-2010 | Assess P datasets for Q simulation. | SWAT, daily, subbasins ~1.4 km² (~1.2 km)². | **P**: CFSR better. **Q**: PRISM better. Note: PRISM data do not appear to be time-shifted. |
| Raimonet et al. (2017) [France] | SAFRAN MESAN E-OBS WFDEI-GPCC | RG DRG G RG | 931 stations, 10-10,000 km², diversity of climate, topo, elev. | 1989-2010 | Evaluate P datasets, daily Q simulation. | GR4J, daily, conceptual, | High-res and reanalysis Q performed better. Essential to account for high-res topo. |
| Ray et al. (2022) [Texas] | Daymet v3 PRISM IMERG-Early v6 IMERG-Late v6 IMERG-Final v6 PERSIANN PERSIANN-CCS PERSIANN-CDR CHIRPS v2 | G G S S SG S S SG SRG | 4,300 km², 111-596 m asl | 2000-2019 | Assess P datasets for Q simulation. | SWAT, daily, subbasins ~50 km² (~7 km)² | Daymet, PRISM, CHIRPS best for Q. |
| Setti et al. (2020) [India] | IMD TMPA-3B42RT TMPA-3B42 NCEP-CFSR | G S SG R | 9,056 km², 152-1600 m asl, 1,140 mm/y P | 1998-2012 | Assess P datasets for Q simulation. | SWAT, daily, subbasins ~211 km² (~14.5 km)² | Good P for all datasets. Q simulation (monthly calibration) good for all (IMD best). |
| Shuai et al. (2022) [Colorado] | PRISM Daymet NLDAS-2 | G G G | 53.2 km² | 2016-2019 (PRISM shift 1 d) | Assess datasets for P (7 sta), T (4 sta), simulated Q, SWE, ET. | ATS, hourly, resolution 0.005-0.05 km², distributed | Small T diff (r>0.95). Strong P corr (r>0.9) for PRISM 93 sites, Daymet (1 site), NLDAS (0 sites). Q (hourly): Daymet > PRISM > NLDAS. |
| Singh & Najafi (2020) [Canada] | NRCan NCEP-CFSR GRASP NCEP-NARR S14FD | G R R RG DR | 113 stations, 3 basins: 46,600 km², 2130-3700 m asl; 600 km², 0-2,000 m asl; 261 km², 0.4-1584 m asl. | 1980-2010 | Assess P, T covariability | Raven, daily, lumped/semi-distributed | Gridded T shows cold bias over Rockies vs warm bias over Prairies. NRCan (and S14FD) best T. |
| Tarek et al. (2020) | EPA-Interim EPA5 | R R | 3,138 basins in US, Canada | 1979-2018 | Evaluate ERA5 vs observations | GR4J, HMETS, | ERA5 improved over ERA-Interim, with biases in SE |





| | | | | | | with emphasis on Q modeling | daily, conceptual | US, W coast of N Amer. Translated into Q skill, except E US. |
|---|---|---|---|---|---|---|---|---|
| Yang et al. (2014) [China] | NCEP-CFSR APHRODITE China-trend | R G G | 2 basins: 1098 km², 366 km² | 2000-2006 | | Calibration, daily Q simulation | SWAT, daily, subbasins ~29 km² (~5.4 km)² | China-trend was best. Poor results in areas with topo influence on P. |
| Zhu et al. (2018) [NE China] | Fengyun TMPA-3B42RT TMPA-3B42 CMORPH-BLD v1 CMORPH v1 | SRG S SG SG S | 12,385 km², 172-1391 m asl, 776 mm/y P | 2006-2010 | | Evaluate five P datasets with gauge P and simulated Q. | SWAT, daily, monthly, subbasins ~459 km² (~21 km)² | Better P agreement from Fengyun, TMPA-3B42, CMORPH-BLD (all gauge-adjusted). Daily Q satisfactory for Fengyun, TMPA-3B42. Model parameters were only applicable for dataset used for calibration. |

**Reference**: [Location]=General region of study. **Data Source**: G=ground-based observations (with interpolation), S=satellite, R=reanalysis, D=downscaling. **Analysis Goals, Hydrologic Outcomes**: NSE=Nash-Sutcliffe efficiency

### 4.1 Humidity, Wind, and Solar Radiation Dataset Assessment

The fewest studies were found that assessed and compared humidity (rh, Tdp, or Vp), windspeed (u), and solar radiation (Rs) to station data or their effects on hydrologic analyses (**Table 4**). Mourtzinis et al. (2017) assessed and compared G gridded datasets for rh (Daymet [derived from Vp], PRISM [derived from Tmin, Tmax]) and Rs (Daymet, NASA-POWER [RS]) to observed data from 45 stations in the midwest U.S. They found good agreement between daily Rs and station data (RMSE=8% for both datasets) with 98% of data within 15% of the measured data. However, daily rh agreement was poor for both Daymet (RMSE=13%) and PRISM (RMSE=18%). Bandaru et al. (2017) compared four gridded datasets (three G, one RG) to observed monthly data from five flux towers in the northwest U.S. and found different results for humidity (Tdp) and Rs. For Tdp, performance decreased from NCEP-NARR (RG) to PRISM to Daymet to NLDAS (G). Conversely for Rs, performance decreased from NLDAS to Daymet (both with negative bias) to NCEP-NARR (positive bias). No studies were found that assessed and compared gridded u datasets.

Current literature does not provide a consensus for humidity, u, or Rs gridded dataset selection. More studies are needed both to assess the accuracy of available humidity, u, or Rs gridded datasets (**Tables 1, 2, 3**) and to assess their impacts on hydrologic model performance. Analyses where rh, u, and Rs are primary forcing variables (e.g., ET, airshed, or surface soil moisture dynamic analyses) may require an assessment of available dataset suitability (e.g., comparison of the gridded dataset to reference, ground-based weather stations in or around the study area) and a sensitivity analysis of the model (how responsive is the response variable to the noted gridded climate dataset uncertainty) prior to dataset selection. Hybrid data sources (station and gridded) need to be considered regarding both model skill for simulating hydrology and optimal model parameter sets, because effects of mixing data sources are generally unknown. Dependencies among climate variables (such as discussed in





Section 4.4 for P-T dependencies) may also be an important consideration for humidity, u, and Rs and lead to prioritizing a
gridded dataset that represents covariances among variables of concern. As such, methods to retain coupling of climate
variables in gridded datasets are needed.

## 4.2 Temperature (T) Dataset Assessment

Accuracy and agreement of gridded datasets of air temperature (T) at 2 m above ground (**Table 4**), about crop canopy height,
were dependent on many factors. Essou et al. (2016b) found T from six reanalysis (R) datasets generally were comparable to
station data in 370 basins across the CONUS. Behnke et al. (2016) evaluated eight G datasets and found gridded T data highly
correlated (r>0.9) with station data, but biased towards cooler T, across the CONUS; the best dataset differed by region, and
spatial resolution was not an important factor. Massmann (2020) analysed three datasets (two R, one G) with long (century)
periods of record in 168 basins throughout the CONUS and found T datasets generally were adequate across the U.S. but were
less adequate in the Rocky Mountains. In the Rockies, Shuai et al. (2022) found strong correlation (r>0.95) with measured
station data in Colorado for G datasets (PRISM, Daymet, NLDAS-2). In the Midwest U.S., Mourtzinis et al. (2017) found
good agreement (RMSE<5%) for both PRISM and Daymet. Tercek et al. (2021) revealed a characteristic of G datasets tending
to underrepresent higher elevation point locations (e.g., mountain tops), which corresponded to gridded monthly maximum T
data. As expected, datasets resulting from downscaling methods were constrained by inherent inaccuracies of the original
gridded T dataset (Gupta & Tarboton, 2016).

Several studies specifically addressed gridded T dataset contribution to hydrologic model performance. A consensus across
many studies was that T dataset selection was less influential on hydrologic simulation accuracy than P dataset selection
(Dembélé et al., 2020; Essou et al., 2016a; Mei et al., 2022; Shuai et al., 2022). Laiti et al. (2018) evaluated five gridded daily
T datasets covering a basin in the Italian Alps, with elevations ranging from 185 to 3,500 m. They found G datasets with higher
resolution produced the best streamflow (Q) simulation, but suggested T datasets from various sources (G, S, R) can be used
interchangeably, with negligible impacts on simulation results. The consensus from these studies suggests that gridded T
datasets generally can be used interchangeably for hydrologic analyses in most parts of the CONUS or globally, but differences
in hydrologic response may arise in areas of more complex (i.e., mountain) topography.

## 4.3 Precipitation (P) Dataset Assessment

Precipitation (P) datasets were less reliable than T datasets both in their accuracy and in their performance forcing hydrologic
models. P data often lack accuracy and spatial variability in complex, mountain topography (Hafzi & Sorman, 2022; Henn et
al., 2018) and need to be gauge corrected (Raimonet et al., 2017; Mazzoleni et al., 2019; Laiti et al., 2018; Essou et al., 2017;
Hafzi & Sorman, 2022). In mountainous regions as well as humid regions, R datasets generally performed better than G in
areas with low station density (< 1 per 1000 km$^2$), but for higher station densities (> 3 per 1000 km$^2$), there was no difference
(Essou et al., 2017). Gampe & Ludwig (2017) and Essou et al. (2016b) found that R datasets show great potential to provide





reasonable P data where station location and density cause high errors and uncertainty, especially in higher elevations and topographically complex regions. Essou et al. (2016b) judged that differences between R and observed G data across the CONUS were small enough to allow direct use of R-based P and T data for hydrologic modelling without bias correction. Satellite datasets corrected with either R or G datasets increased P accuracy (Hafzi & Sorman, 2022). In hydrologic models, inputs of daily or hourly P are partitioned into rain (liquid) and snow (solid) based primarily on T (daily minimum T), but little

information exists on the relative accuracy of P for rain, snow, and rain/snow mixes. See Sections 4.4 and 4.6 for P-T interactions and estimation of snow-water equivalent (SWE).

The G methods provided the most accurate gridded P data (Kouakou et al., 2023; Massmann, 2020), although performance of G datasets deteriorate in gauge-sparse regions (Beck et al., 2017b). With adequate station density, weather-station network data were superior at local to regional scales (Tarek et al., 2020; Meng et al., 2014; Yang et al., 2014). Datasets that directly

integrated higher temporal resolution gauge data performed best, with decreasing performance from those incorporating daily gauge data (CPC-Unified, MSWEP v1.2 and v2) compared to 5-day gauge data (CHIRPS v2), monthly gauge data (GPCP-1DD v1.2, TMPA-3B42 v7, WFDEI-CRU), or monthly SG GPCP product (PERSIANN) (Beck et al., 2017b).

Global R datasets often were good proxies for P data (Essou et al., 2016b). Massmann (2020) found R datasets were more appropriate for short-term P in the northwest U.S., with some difficulties in representing P in the south and east U.S. From a

comparison of 18 gridded datasets, Mazzoleni et al. (2019) found no single best P dataset. The R datasets performed better than G datasets for low station density ($<1$ per 1000 km$^2$); otherwise, little difference was observed (Essou et al., 2017; Tarek et al., 2020). Gampe & Ludwig (2017) found higher resolution P data performed better, but coarse data provided close representation of overall, longer-term climate characteristics. Raimonet et al. (2017) demonstrated the importance of accounting for the impacts of high-resolution topography on P gridded data, and that low-altitude, less-complex topographies were less sensitive to the choice of gridded dataset. Similar results were reported by Laiti et al. (2018), who added that simple

bias correction cannot overcome P dataset deficiencies.

### 4.4 P-T Dependency

Climatological dependencies can exist between P and T. Gridded datasets decouple P and T, which can cause problems with hydrologic simulation (Singh & Najafi, 2020). For example, Singh & Najafi (2020) noted failure of gridded datasets to

represent warm-wet dependencies in north and southwest Canada and hot-dry dependencies in spring and summer seasons in the Canadian prairies that were present in the observed data. This led to inaccurate modelling of hydrologic processes (rain/snow partitioning, extreme events), which may be particularly important in representing hydrological reality under a changing climate. In response to this need for coupled P and T data, Raimonet et al. (2017) suggested a process of dynamically calibrating a conceptual hydrological model on meteorological datasets, which was able to assess consistency of the

meteorological datasets, including covariance of P and T, as well as improve streamflow simulation performance. Again, these results suggest that methods to retain coupling of climate variables in gridded datasets are needed.





## 4.5 Streamflow (Q) Modelling

Not surprisingly, as noted above, Q was more responsive to P than T (Dembélé et al., 2020; Essou et al., 2016a; Mei et al., 2022; Shuai et al., 2022). Most gridded P datasets were adequate for Q simulation at the monthly scale (Ray et al., 2022; Meng

et al., 2014; Muche et al., 2020; Setti et al., 2020). Some studies found that G-based P datasets generally were better than S or R datasets for hydrological modelling (Ray et al., 2022; Meng et al., 2014; Kouakou et al., 2023; Massmann, 2020), especially for high spatial resolution datasets (Laiti et al. 2018). In addition, hydrologic performance using G datasets was not affected by basin size, but that performance of the G datasets did improve slightly as the weather station density of their source data increased (Essou et al., 2017). However, total basin size did not influence Q performance (Tarek et al., 2020). In a study of 8

large-scale basins globally, Mazzoleni et al. (2019) found Q simulation was affected by basin scale, human footprint, and climate: S datasets had the poorest performance and were the most variable; SG datasets were the best performers in tropical and temperate-arid climates; and RG datasets were the best performers in temperate and temperate-cold climates and within densely gauged P basins.

In a study of 9 gridded datasets applied with a conceptual hydrologic model to simulate streamflow in 9,053 basins (<50,000

km$^2$) worldwide, Beck et al. (2017b) found MSWEP v2 P dataset provided consistently better performance than other products across North America, Europe, Japan, Australia, New Zealand, and southern and western Brazil, whereas CHIRPS v2 performed better than other products in Central America, and central and eastern Brazil, but no one dataset performed best everywhere. They also concluded, based on good performance of CPC-United, CHIRPS v2, and MSWEP v1.2 and v2, that incorporation of sub-monthly gauge data improved Q simulation.

Interestingly, the best P dataset was not always the best for Q modelling (Yang et al., 2014), and lower P and T performance did not always translate into lower Q performance (Essou et al., 2016a; Hafzi & Sorman, 2022). Similarly, Ang et al. (2022) found that S datasets corrected with G observations had better Q performance than other G or R datasets that performed similarly in comparison to observed P data.

Datasets with the best representation of temporal dynamics did not necessarily align with those with the best representation of

spatial patterns, with more hydrologic uncertainty associated with misrepresenting spatial patterns than temporal dynamics (Dembélé et al., 2020). Hafzi & Sorman (2022) found most gridded P datasets had low performance in detecting daily P over space and time, but some still had accurate Q simulation. Mazzoleni et al. (2019) found that the best P dataset for a basin outlet was not necessarily the best for its subbasins, which reflects the influence of scale and suggests a benefit to distributed hydrologic modelling over lumped modelling approaches.

Hydrologic model calibration approaches were also sensitive to selection of gridded dataset. Ray et al. (2022) found that model parameter uncertainty decreased when calibrating the SWAT model using G-based P datasets. In addition, hydrologic models calibrated using one gridded dataset did not work as well when applied using forcings from other datasets (Zhu et al., 2018; Hafzi & Sorman, 2022).





Dependency between P and T did not appear to affect Q simulation. Shuai et al. (2022) found that inter-mixing T datasets
among PRISM, Daymet, and NLDAS P datasets had little effect on Q.

### 4.6 ET and SWE Modelling

Few studies were found that compared the effects of gridded datasets on simulation of other spatially distributed hydrologic
variables, such as ET or SWE. Mourtzinis et al. (2017) found Daymet outperformed PRISM in calculating FAO-Penman-
Monteith reference ET (ETo) in the midwest U.S. Although both were similar in comparisons of P and T, ETo bias was less
for Daymet (-4 mm) than PRISM (+253 mm), and both had poor agreement in the high and low ranges of measured ETo.
Errors were related to poor agreement with rh, especially for PRISM. Shuai et al. (2022) found little difference between
simulation of ET from PRISM, Daymet, and NLDAS in Colorado and assumed the similarity was related to using the same
Rs forcing. Shuai et al. (2022) also evaluated the effects of G datasets on SWE in Colorado. They found fine spatial scale
helped PRISM (0.8 km) and Daymet (1 km) outperform NLDAS (12 km) in simulating spatial SWE, with the highest
correlation from PRISM. Gupta & Tarboton (2016) used spatially downscaled R datasets (MERRA data for T, rh, u, and Rs;
RFE v2 data for P) and found good SWE simulation compared to SNOTEL data (mean NSE=0.67 across 8 sites). Key sources
of discrepancies were from P and Rs data uncertainty.

### 4.7 P Ensembles

Ensembles of gridded datasets often were recommended to account for gridded dataset uncertainty and better represent overall
climatology (Gampe & Ludwig, 2017; Pokorny et al., 2020), but with some caveats. For example, Gampe & Ludwig (2017)
found R data (compared to station data) showed fewer consecutive dry days (CDDs, P < 1 mm), more consecutive wet days
(CWDs, P > 1 mm), and lower contribution of heavy P events (i.e., more low but steady P events) to annual P, which has the
potential to impact hydrologic simulation (more infiltration, less streamflow, greater baseflow, fewer floods, etc.). They
recommended identification and removal of such non-representative datasets from ensembles. Pokorny et al. (2020) suggested
that data should be assessed in relation to the target hydrologic model's spatio-temporal scale. Notably, ensembles dampen
extreme events and decrease the frequency of low/high P events, which can lead to non-representative hydrological simulation
(Pokorny et al., 2020). Laiti et al. (2018) demonstrated a Hydrologic Coherence Test (HyCoT) to exclude meteorological data
based on modelling goal.

### 4.8 Latency

The latency with which gridded datasets become available for use may be a critical factor in gridded dataset selection. Few
studies assessed latency effects. Hafzi & Sorman (2022) found that some real-time datasets that were available with short
latency (e.g., 1-h lag, PERSIANN-CSS with 0.04°) sacrificed accuracy compared to coarser, longer-latency datasets, such as
IMERG-Late v6 (14-h lag, 0.1°), MSWEP v2.8 (few-month lag, 0.1°) and CHIRPS v2 (1-month lag, 0.05°).





**5 Conclusions**

This study summarized characteristics, primary references, and data availability of 60 gridded datasets at CONUS to global extents to assist in dataset selection by hydrologic investigators. A review of information from 28 recent (past 10 years) intercomparison studies spans a wide range of gridded datasets, study settings and scales, and hydrologic modelling objectives. Readers are referred to these studies for a wealth of details on their results and recommendations; we encourage particular focus on studies with similar climatic setting and hydrologic objectives. Herein we strived to describe and synthesize the key

lessons learned.

  No single gridded climate dataset or data source was universally superior for hydrologic analyses. Several common themes arose among the 28 studies reviewed. Gridded daily temperature datasets improved when derived from greater station density, though they were relatively interchangeable in hydrologic analyses. Gridded daily precipitation data were more accurate when derived from higher-density station data, when used in spatially less-complex terrain, and when corrected using ground-based

data. In mountainous or humid regions, reanalysis-based gridded datasets generally performed better than ground-based gridded datasets when the underlying station density was low; but when station densities were higher, there was no difference. Ground-based gridded precipitation datasets generally performed better than satellite- or reanalysis-based datasets, though better precipitation and temperature datasets did not always translate into better streamflow modelling. Hydrologic analyses would benefit from advances in creating gridded datasets that retain climate variable interdependencies and better represent

climate variables in complex topography. The caveat that some studies were insensitive to using independent sources of P and T may not be a good rationale for ignoring possible cross-correlations between climate variables. Rather, this result may point to the insensitivity of hydrologic models that don't necessarily capture space-time process interactions within a watershed. Use of hybrids of gridded datasets and station data for a particular region remains a topic for further investigation because there can be substantial differences between data at a particular station and its corresponding grid-cell data.

Hydrologic investigators should justify their selection of a particular gridded dataset with full consideration of both the climatologic setting and the hydrologic analysis type and objectives. This study provides some general consensus recommendations, though characteristics of a given hydrologic analysis or study may warrant more specific selection processes and criteria.

**Appendix**

**Table A1. Summary of Different Data Formats, Descriptions, and Processing Approaches for Gridded Climate Datasets using Programming Languages and Software**

| Data Name | Format Description |
| --- | --- |





| | |
|---|---|
| **NetCDF**<br>(Network<br>Common Data<br>Form) | The NetCDF format was first developed in the 1980s by researchers at the Unidata Program Center at the University Corporation for Atmospheric Research (UCAR). Since then, it has undergone several revisions and updates to address technology and user needs changes. The latest version, NetCDF-4, includes support for compression, chunking, and parallel I/O, as well as new data types and features for handling large and complex datasets. NetCDF files are widely used in the atmospheric and climate science communities and are supported by many software packages. They include metadata that describe the file's contents and allow easy data access. NetCDF is a self-describing format, meaning the metadata are embedded within the file. This makes sharing and using the data more accessible, as the metadata travel with the data. NetCDF files can be read and written using a variety of software packages, including Python, R, and MATLAB.<br><br>The NetCDF format can be accessed and manipulated using a variety of software packages, including:<br><br>● NetCDF software library: A library of programming functions for working with NetCDF files in C, Fortran, and other programming languages.<br><br>● NetCDF4-Python: A Python package that provides access to NetCDF files using the NetCDF-4 library.<br><br>● RNetCDF: An R package that provides access to NetCDF files using the NetCDF library.<br><br>● Panoply: A Java-based application for visualizing and analyzing NetCDF files.<br><br>Some tutorials on working with NetCDF files include:<br><br>● Unidata NetCDF tutorials: http://www.unidata.ucar.edu/software/netcdf/docs/netcdf-tutorial/<br><br>● Python NetCDF4 tutorial: https://unidata.github.io/netcdf4-python/netCDF4/index.html<br><br>● RNetCDF tutorial: https://www.r-bloggers.com/2019/08/working-with-netcdf-files-in-r/ |
| **HDF5**<br>(Hierarchical<br>Data Format 5) | The HDF5 format was first introduced in 1997 by the National Center for Supercomputing Applications (NCSA) at the University of Illinois at Urbana-Champaign. Since then, it has become a widely used format for scientific data, including climate data. HDF5 has undergone several revisions and updates, including supporting compression, chunking, parallel I/O, and new features for managing large and complex datasets. HDF5 is a flexible and efficient format that can handle various data types, including climate data. It includes features for managing large and complex datasets, such as compression, chunking, and parallel I/O. HDF5 files are portable across platforms and can be accessed using a variety of programming languages, including Python, R, and MATLAB. However, HDF5 can be more complex to work with than other formats, and the metadata are not always embedded within the file itself, making it harder to share and use the data.<br><br>The HDF5 format can be accessed and manipulated using a variety of software packages, including:<br><br>● HDF5 software library: A library of programming functions for working with HDF5 files in C, C++, Fortran, and other programming languages.<br><br>● h5py: A Python package that provides access to HDF5 files using the HDF5 library.<br><br>● rhdf5: An R package that provides access to HDF5 files using the HDF5 library.<br><br>● HDF Compass: A graphical tool for exploring and editing HDF5 files.<br><br>Some tutorials on working with HDF files include:<br><br>● HDF Group HDF5 tutorial: https://portal.hdfgroup.org/display/HDF5/Tutorials<br><br>● Python h5py tutorial: https://www.h5py.org/docs/ |





| | |
|---|---|
| | ● R hdf5r tutorial: https://cran.r-project.org/web/packages/hdf5r/vignettes/hdf5r.pdf |
| **ASCII** (American Standard Code for Information Interchange) | ASCII is a simple text format that has been in use for decades. While there have been no significant changes to the format, technological advances have made working with large datasets in ASCII format easier. ASCII files are easy to read and write but can be less efficient for storing large datasets. ASCII files can be opened and edited using any text editor, but additional processing may be required in other software packages. Some tutorials to handle simple text files include: <br>● Python CSV tutorial: https://realpython.com/python-csv/ <br>● R readr tutorial: https://readr.tidyverse.org/articles/readr.html <br>● MATLAB import data function documentation: https://www.mathworks.com/help/matlab/ref/importdata.html |
| **GRIB** (Gridded Binary) | The GRIB format was first introduced in the 1980s by the World Meteorological Organization (WMO) to standardize the exchange of weather and climate data. Since then, it has undergone several revisions and updates to address technology and user needs changes. The latest version, GRIB2, includes support for new data types and features for encoding and compressing data, which can make them more compact than other formats. However, GRIB files can be more complex than other formats and may require specialized software to read and write. <br>The GRIB format can be accessed and manipulated using a variety of software packages, including: <br>● ECMWF GRIB API: A software library for working with GRIB files developed by the European Centre for Medium-Range Weather Forecasts (ECMWF). <br>● PyGRIB: A Python package that provides access to GRIB files using the ECMWF GRIB API. <br>● R package "gribtools": Provides tools to manipulate, read and write GRIB files. <br>● wgrib2: A command-line tool for manipulating and converting GRIB files. <br>Some tutorials on working with GRIB files include: <br>● ECMWF GRIB API tutorial: https://software.ecmwf.int/wiki/display/GRIB/GRIB+API+tutorial <br>● Python PyGRIB tutorial: https://jswhit.github.io/pygrib/docs/pygrib.html <br>● R rNOMADS tutorial: https://cran.r-project.org/web/packages/rNOMADS/vignettes/rNOMADS.html |
| **GeoTIFF** (Georeferenced Tagged Image File Format) | The GeoTIFF format was first introduced in the 1990s to include georeferencing information in TIFF image files. Since then, it has become a widely used format for storing and analyzing spatial data, including climate data. GeoTIFF has undergone several revisions and updates, including the addition of support for new coordinate systems and projections and new features for managing large and complex datasets. GeoTIFF files include spatial information, making them useful for storing and analyzing climate data that are geographically referenced. GIS software packages widely support them and include metadata describing the coordinate system, projection, and other data attributes. However, GeoTIFF files can be larger than other formats and may require specialized software to read and write. <br>The GeoTIFF format can be accessed and manipulated using a variety of software packages, including: <br>● GDAL (Geospatial Data Abstraction Library): A software library for reading and writing geospatial data, including GeoTIFF files. GDAL can be accessed using Python, R, and other programming languages. <br>● R package "raster": Provides tools to manipulate, read, and write GeoTIFF files in R. |





- QGIS: A free and open-source GIS software package that includes tools for working with GeoTIFF files.
- ArcGIS: A proprietary GIS software package that includes tools for working with GeoTIFF files.

Some tutorials to help understand working with GeoTIFF files include:

- GDAL/OGR tutorial: https://gdal.org/tutorials/raster_api_tut.html
- Python rasterio tutorial: https://rasterio.readthedocs.io/en/latest/topics/index.html
- QGIS training manual: https://docs.qgis.org/3.16/en/docs/training_manual/index.html

## Data Availability

The data that support the findings of this study are available from the corresponding author upon reasonable request.

**Author contribution**

KM conceptualized the study and prepared original draft. SM compiled the data and contributed to preparing the original draft, KM and SM developed methodologies and interpreted data. TG and DB provided manuscript critical review and revisions.

## Competing Interests

The authors declare they have no conflict of interest.

**Acknowledgment**

USDA is an equal opportunity employer and provider. This research was supported by the USDA, Agricultural Research Service. The findings and conclusions in this publication are those of the author(s) and should not be construed to represent any official USDA or U.S. Government determination or policy. Mention of trade names or commercial products in this publication is solely for the purpose of providing specific information and does not imply recommendation or endorsement by
the USDA.

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
