# Peer review of "Review of Gridded Climate Products and Their Use in Hydrological Analyses Reveals Overlaps, Gaps, and Need for More Objective Approach to Model Forcings"

_Hydrology and Earth System Sciences, 2024_

## Referee Comment (RC2)

This manuscript compiles many meteorological forcings datasets and provides an overview, which has certain reference significance for modeling research. However, I think the narrative in the article needs to be further improved.

**Major Comments:**
**Introduction.** The descriptions are too simplistic in this manuscript. (e.g., "Many studies have intercompared the accuracy of particular subsets of these gridded climate datasets for various regions, settings, and time frames across the globe with various insights and conclusions."). More citations are needed to support your opinion and illustrate with specific examples.

**Minor Comments:**
**Abstract.** The manuscript can summarize the advantages of this work, for example, including the situation of previous research, and the innovation of this research.

**Figure 1.** This picture needs further beautification. In addition, some explanation should be added in the title of the figure, Such as "Spatial Coverage: Land=Global land surfaces only (not ocean surfaces).".

**Section 3.3.** The spatial and temporal resolution of evapotranspiration, runoff, and other hydrological elements is relatively high (100m-1km, hourly; Melsen et al., 2016). The resolution of gridded climate datasets should be an important criterion to consider. In my opinion, the resolution of the hydrologic model is limited by the spatial and temporal resolution of climate datasets. Hence, the manuscript should clarify the significance of high-resolution gridded climate datasets for numerical simulation, especially for reducing uncertainty in simulation.

Reference: Melsen LA, Teuling AJ, Torfs PJJF, et al. HESS Opinions: The need for process-based evaluation of large-domain hyper-resolution models[J]. Hydrology and Earth System Sciences, 2016, 20(3): 1069-1079.

---

## Author Response (AR1)

**Reviewer Comments: RC1: 'Comment on hess-2024-58', Anonymous Referee #1, 22 Apr 2024**

This compilation of gridded climate datasets will likely be a useful resource for hydrologic modelers in selecting an appropriate data product. The authors conducted an extensive search for relevant products and summarize them, as well as studies that have conducted dataset comparisons in several tables. This meta-analysis again is a good resource but as it is currently presented is a list of resources with some explanatory information, rather than a true synthesis. Additional examples, citations, and synthesis of the dataset comparison papers would make a huge improvement to this manuscript. I recognize that each study site, modeling goal, and additional constraints mean that the final selection of datasets for an individual project will be variable and therefore the authors cannot reasonably make specific recommendations. However, I think readers could benefit from additional synthesis and examples so they could identify parallels in their own work and make better informed decisions on dataset choice, particularly in all subsections of section 3. I suggest the authors add more specific examples with citations to these paragraphs. Similarly, all parts of section 4 read like the authors are rattling off a list of findings from each of the studies rather than synthesizing them into a more cohesive narrative.

RESPONSE: The reviewer identifies the key limitation (and frustration!) of this study: "…each study site, modeling goal, and additional constraints mean that the final selection of datasets for an individual project will be variable and therefore the authors cannot reasonably make specific recommendations". Additional references/examples would also be subject to this limitation. We have tried to add more specifics, but the comparison methodologies employed by many of the studies are complex, and each one is different, such that multiple sentences would be needed to clarify each result, which would greatly increase the size of the paper and perhaps obfuscate the conclusions/recommendations. We welcome specific recommendations for improvement. Meanwhile, we have tried to enhance the paper more generally to address this identified limitation. Nonetheless, in response to this RC1 reviewer comment and a related RC2 reviewer comment, we tried to steer readers in the right direction by adding the following summary recommendation to both the Abstract and Conclusions: "Based on this study, the authors' overall recommendation is to select the gridded dataset (from Tables 1, 2, and 3) (a) having spatial and temporal resolutions that match modelling scales, (b) that are primarily (G) or secondarily (SG, RG) derived from ground-based observations, (c) with sufficient spatial and temporal coverage for the analysis, (d) with adequate latency for analysis objectives, and (e) that includes all climate variables of interest, so as to better represent interdependencies."

Additionally, I found no discussion of how the dataset assessment studies were found and evaluated to be included in this manuscript.

RESPONSE: The first paragraph of Section 4 describes the criteria used to include studies in the review. The search effort was considerable by both the first and second authors, with all authors contributing literature, and was ended when search terms identified redundant literature.

Minor Comments:

Line 22 of the abstract: confusing as written datasets of what? Temperature? How does this relate to the following sentence?

RESPONSE: Revised: "In mountainous regions as well as humid regions, reanalysis-based precipitation datasets generally performed better than ground-based when underlying data had low station density, but for higher station densities, there was no difference. Ground-based  datasets generally  were more accurate in representing precipitation and temperature data than satellite- or reanalysis-based datasets, though  this did not always translate into better streamflow modelling."

Line 34, need some citations

RESPONSE: We added some specific text to clarify the statement (see underlined text), and note Section 4 (Table 4), which provides citations and details of 28 recent studies that support this statement: "Many studies (28 of which are reviewed in Section 4 of this article) have intercompared… A search of "intercomparison" AND "gridded AND climate AND data" yielded 202 documents using Scopus. Excluding "climate change" reduced this to 100 documents, and excluding "CMIP" produced 77 documents.".

The introduction is very short, which I think is fine for this manuscript, but I do suggest that some of the more introductory information in section 2 (particularly the first paragraph) be moved to the introduction to make a slightly more comprehensive introduction.

RESPONSE: Rearranged as suggested. First paragraph of Section 1 was split, and the first paragraph of Section 2 was inserted.

Line 97 – more direct, don't need hyphen.

RESPONSE: APA Style Guide: "In a temporary compound that is used as an adjective before a noun, use a hyphen if the term can be misread or if the term expresses a single thought (i.e., all words together modify the noun)." This doesn't seem to be easily misread, but the two terms do not independently modify the noun ("more measure" doesn't make any sense without "direct"). Since the APA rule is "or", a hyphen would be indicated, though I have no problem omitting it, at the editor's discretion.

Figure 1- This figure is very poorly made and, in my opinion, not of publication quality.

RESPONSE: Agreed. Figure 1 was upgraded.

Line 210 – Latency should be defined much earlier in the manuscript than here.

RESPONSE: In the first paragraph of Section 1 (now the third paragraph), we briefly defined the latency term: "…gridded datasets often are not available in real-time (i.e., data latency)…". Is this sufficient?

Line 236 – The fewest? Can this be supported with a % or n?

RESPONSE: Yes, thanks. The prior version of Figure 4 showed this explicitly. Clarification added: "(only 1 of the 28 studies in Table 4)".

Line 248-249, I would include snowpack in this list

RESPONSE: Agreed, added.

**Reviewer Comments: RC2: 'Comment on hess-2024-58', Anonymous Referee #2, 21 May 2024**

This manuscript compiles many meteorological forcings datasets and provides an overview, which has certain reference significance for modeling research. However, I think the narrative in the article needs to be further improved.

Major Comments:

Introduction. The descriptions are too simplistic in this manuscript. (e.g., "Many studies have intercompared the accuracy of particular subsets of these gridded climate datasets for various regions, settings, and time frames across the globe with various insights and conclusions."). More citations are needed to support your opinion and illustrate with specific examples.

RESPONSE: The Introduction was reorganized and revised in response to this comment and another from Reviewer 1. For the example noted above, we added some specific text to clarify the statement (see underlined text), and note Section 4 (Table 4), which provides citations and details of 28 recent studies that support this statement: "Many studies (28 of which are reviewed in Section 4 of this article) have intercompared the accuracy of particular subsets of these gridded climate datasets for various regions, settings, and time frames across the globe with various insights and conclusions.  A search of "intercomparison" AND "gridded AND climate AND data" yielded 202 documents using Scopus. Excluding "climate change" reduced this to 100 documents, and excluding "CMIP" produced 77 documents. Even with these filters, most studies focus on a limited number of datasets, lack generalizable recommendations, and do not consider the functional implications of dataset limitations on end-users' hydrologic analysis. The present study aims to provide a comprehensive compilation, overview, and considerations for selection of gridded datasets with focus on selection for hydrologic modelling and analyses. Our focus is on historical datasets (not climate projections) at the conterminous U.S. (CONUS) to global extents."

Minor Comments:

Abstract. The manuscript can summarize the advantages of this work, for example, including the situation of previous research, and the innovation of this research.

RESPONSE: The following summary recommendation was added to both the Abstract and Conclusions: "Based on this study, the authors' overall recommendation is to select the gridded dataset (from Tables 1, 2, and 3) (a) having spatial and temporal resolutions that match modelling scales, (b) that are primarily (G) or secondarily (SG, RG) derived from ground-based observations, (c) with sufficient spatial and temporal coverage for the analysis, (d) with adequate latency for analysis objectives, and (e) that includes all climate variables of interest, so as to better represent interdependencies."

Figure 1. This picture needs further beautification. In addition, some explanation should be added in the title of the figure, Such as "Spatial Coverage: Land=Global land surfaces only (not ocean surfaces).".

RESPONSE: Agreed. Figure 1 was upgraded, and figure caption was expanded to define abbreviations.

Section 3.3. The spatial and temporal resolution of evapotranspiration, runoff, and other hydrological elements is relatively high (100m-1km, hourly; Melsen et al., 2016). The resolution of gridded climate datasets should be an important criterion to consider. In my opinion, the resolution of the hydrologic model is limited by the spatial and temporal resolution of climate datasets. Hence, the manuscript should clarify the significance of high-resolution gridded climate datasets for numerical simulation, especially for reducing uncertainty in simulation.

RESPONSE: Thank you, nice comment. We expanded the text, accordingly: "The simulated spatial and temporal resolution of evapotranspiration (ET), runoff, and other hydrological elements in hydrologic models can be relatively fine (<1 km, subdaily), and model resolution is increasing in ways that capitalize on increasing computational power, process understanding, and data availability (Melsen et al., 2016). Hydrologic model output resolution and uncertainty are often limited by the spatial and temporal resolution of climate datasets. As such, the resolution of gridded climate datasets should be an important criterion to consider."

Reference: Melsen LA, Teuling AJ, Torfs PJJF, et al. HESS Opinions: The need for process-based evaluation of large-domain hyper-resolution models[J]. Hydrology and Earth System Sciences, 2016, 20(3): 1069-1079.

---

## Referee Report (RR1)

Major comments

1. The authors provide a valuable service to the hydrologic modeling community. Firstly, simply listing the major differentiating features of these spatial climatic datasets can assist researchers in evaluating their own methods (for example, I was not aware that there was such a degree of variability in latency). Secondly, this summary of available modern datasets can facilitate an improved dataset selection and justification process in future studies.
2. At several points the manuscript resembles a laundry list of findings derived from a review of data sources or other studies. This is partly unavoidable due to the nature of the analysis, but the authors can assist a reader by inserting additional summary text before and/or after the paragraphs in which the resources or findings are listed (see minor comments for line 274, 291, and 295).
3. As a reader I found the use of example research applications very grounding, since much of the paper is, by necessity, composed of high-level summarizing. I have noted some places where I think additional concrete examples or context would be useful.

Minor comments

| Line No. | Comment |
|---|---|
| 93 | Make sure column headers are included on each page where Table 1 appears. |
| 119 | Make sure column headers are included on each page where Table 2 appears. |
| 130 | Rephrase with semicolons: "Reanalysis systems use various models, observational datasets, and assimilation methods; can generate many climate variables with inter-dependent variable consistency; and provide near-real-time datasets with latency periods from hours to months." |
| 140 | Make sure column headers are included on each page where Table 3 appears. |
| 149 | "merging other data sources" - consider specifying that they are other data sources with complimentary advantages or disadvantages. |
| 152 | How do they estimate accuracy and bias (i.e., do they train the predictions on a training set and leave out certain ground stations or satellite observations as a test set for calculating error)? |
| 164 | Consider rephrasing: "Many characteristics of gridded datasets influence the best product for a given application or research question." |
| 175 | In this paragraph, consider adding a sentence describing what fraction of datasets include a single variable (e.g., P) versus multiple variables. |
| 186 | Consider adding another example: calculations of the recurrence interval of a flood of a given severity. |

| Line No. | Comment |
|---|---|
| 218-220 | I find these example research applications helpful and would recommend adding at least one of to each of the subsections in Section 3, Considerations for Use. Alternatively, you could put several different example applications and what features they might prioritize somewhere earlier in Section 3. |
| 247 | Make sure column headers are included on each page where Table 4 appears. |
| 274 | This paragraph is a bit of a laundry list of different findings. To some extent this is unavoidable with this type of review work, but additional summarizing could help the reader. Consider this revision of the topic sentence: "Accuracy and agreement of gridded datasets of air temperature (T) at 2 m above ground (Table 4), about crop canopy height, were dependent on many factors, including spatial region of interest and topography." |
| 279 | Does the author define "adequate"? |
| 291 | This summary statement is helpful. Would recommend adding the number of studies ("consensus from these X studies") to this sentence. |
| 295 | The laundry list problem again - consider adding another summary statement to this section (at the top or bottom). One option would be to frame it from the perspective of a reader looking for the best dataset for their application (i.e., factors X, Y and Z have the biggest impact on dataset accuracy/are most important when choosing a data product). |
| 335 | Adequate for Q simulation at what river scale? |
| 357 | "detecting" - unclear word choice. Possibly mean representing? |
| 368 | Could use another summary statement for this section. |
| 388 | The Hydrologic Coherence Test sounds interesting! Consider including a brief example of how it would be used. |
| 400 | "similar climatic setting and hydrologic objectives" - similar to what? Similar to the reader's own research project? Or is this a recommendation to pay more attention to studies that have higher similarity within the datasets they consider? |
| 420 | This is a long and useful list of considerations. Consider adding "especially in areas of high topographic relief" after "derived from ground-based observations". |
| 425 | Consider adding vertical lines to Table A1 to visually separate columns. The word "Data" in "Network Common Data Form" runs into the descriptive text to its right. |

---

## Author Response (AR2)

**hess-2024-58**

Title: Review of Gridded Climate Products and Their Use in Hydrological Analyses  Reveals Overlaps, Gaps, and Need for More Objective Approach to Model Forcings

Author(s): Kyle R. Mankin et al.

MS type: Review article

Iteration: Revision

**Authors' Response:**

All editor and reviewer comments are addressed below. In addition, all references were reformatted according to journal guidelines.

**Reviewer Comments: RC1: 'Comment on hess-2024-58', Anonymous Referee #1, 09 Sep 2024**

The authors have made minor changes to the manuscript, which while they do improve certain aspects, I think the edits were not substantial enough to make this a considerably more impactful paper. Figure 1 is now a publishable figure and the addition of recommendations in the abstract and conclusion is useful. However, I still believe that the authors could have made an effort to add synthesis and summary in section 4. Many paragraphs are a series of sentences describing the findings of previous studies and include no original interpretation. This paper can be published in its present state, but in my opinion its value and contribution to hydrologic sciences is somewhat limited.

RESPONSE: We reviewed the synthesis in section 4 in response to this comment, and we disagree that there is no original interpretation. The authors of individual papers summarized study results toward the objectives of their individual studies, but further interpretation was often needed to reframe and synthesize the results in a manner that contributed to the objectives of this paper. Many of the summarized results in this paper are not just restatements of prior results.

Probably the most meaningful response we can provide to the "somewhat limited …. value and contribution" comment is the following. In the short time that the preprint has been online, we have already received a positive reader comment:

"I wanted to thank you because it was really helpful to me and concisely summarized the findings of a lot of literature, and pointed me to some studies that I was not previously aware of."

This directly algins with the objective of this paper. It appears that in this regard, the paper is already being successful.

This reader also provided 3 references for our consideration (2 of which were relevant and added in the latest revision [tables 1, 3, and 4, fig 1, and sections 4.1, 4.2, and 4.6]):

"…if it's not too late to add, and I realized it probably is, I found a few more papers that you might want to consider citing. You correctly noted that in the paper that there's not many studies validating gridded humidity datasets, but I did find a few that you didn't cite, but that have been very helpful to me in choosing which humidity datasets might work best in different situations."

Minor comments:

In the abstract, you define R and G early. Then you keep redefining it in parentheses throughout the rest of the abstract. Either just use R and G later in the text or delete it altogether.

RESPONSE: Abstract revised to use R and G.

First sentence in section 4.1: fewer implies plural. I suggest you just lead with "One of the 28 studies" instead of putting it in parentheses at the end of the sentence. This will make your point more succinct and impactful.

RESPONSE: Reworded, as suggested: "Two of the 29 studies summarized in Table 4 assessed and compared…".

**Reviewer Comments: RC3: 'Comment on hess-2024-58', Anonymous Referee #3, 10 Aug 2024**

Minor revisions - add additional summarizing text and concrete examples at various points to guide the reader.

Referee Report: hess-2024-58-referee-report.pdf.

Major comments

1. The authors provide a valuable service to the hydrologic modeling community. Firstly, simply listing the major differentiating features of these spatial climatic datasets can assist researchers in evaluating their own methods (for example, I was not aware that there was such a degree of variability in latency). Secondly, this summary of available modern datasets can facilitate an improved dataset selection and justification process in future studies.

2. At several points the manuscript resembles a laundry list of findings derived from a review of data sources or other studies. This is partly unavoidable due to the nature of the analysis, but the authors can assist a reader by inserting additional summary text before and/or after the paragraphs in which the resources or findings are listed (see minor comments for line 274, 291, and 295).

3. As a reader I found the use of example research applications very grounding, since much of the paper is, by necessity, composed of high-level summarizing. I have noted some places where I think additional concrete examples or context would be useful.

RESPONSE: These overarching comments are addressed specifically below.

Minor comments

Line No. Comment

Make sure column headers are included on each page where Table 1 appears.

RESPONSE: Agreed. I am certain this will be addressed in the final proof editing process.

Make sure column headers are included on each page where Table 2 appears.

RESPONSE: Agreed. I am certain this will be addressed in the final proof editing process.

Rephrase with semicolons: "Reanalysis systems use various models, observational datasets, and assimilation methods; can generate many climate variables with inter-dependent variable consistency; and provide near-real-time datasets with latency periods from hours to months."

RESPONSE: Edited as suggested.

Make sure column headers are included on each page where Table 3 appears.

RESPONSE: Agreed. I am certain this will be addressed in the final proof editing process.

"merging other data sources" - consider specifying that they are other data sources with complimentary advantages or disadvantages.

RESPONSE: Nice addition. Added, with slight revision: "…other data sources with complimentary advantages to reduce errors or biases".

How do they estimate accuracy and bias (i.e., do they train the predictions on a training set and leave out certain ground stations or satellite observations as a test set for calculating error)?

RESPONSE: Each dataset handles this somewhat differently, so this general statement is a marker for the reader to investigate any selected dataset. Tried to address this with the following: "Each dataset follows a different workflow in developing the integrated product; in general, the primary method (in this article, the first abbreviation letter) is enhanced somewhat sequentially with various interpolation or bias-correction schemes using the secondary dataset(s)."

Consider rephrasing: "Many characteristics of gridded datasets influence the best product for a given application or research question."

RESPONSE: Edited as suggested.

In this paragraph, consider adding a sentence describing what fraction of datasets include a single variable (e.g., P) versus multiple variables.

RESPONSE: Unless I'm reading this comment incorrectly, this is already answered in the third column of Tables 1, 2, and 3, where the specific variables in each gridded dataset are listed.

Consider adding another example: calculations of the recurrence interval of a flood of a given severity.

RESPONSE: Nice. Added, as suggested.

218-220 I find these example research applications helpful and would recommend adding at least one of to each of the subsections in Section 3, Considerations for Use. Alternatively, you could put several different example applications and what features they might prioritize somewhere earlier in Section 3.

RESPONSE: All 3.x subsections already had some number of practical examples, and these were augmented judiciously with new examples or applications.

Make sure column headers are included on each page where Table 4 appears.

RESPONSE: Agreed. I am certain this will be addressed in the final proof editing process.

This paragraph is a bit of a laundry list of different findings. To some extent this is unavoidable with this type of review work, but additional summarizing could help the reader. Consider this revision of the topic sentence: "Accuracy and agreement of gridded datasets of air temperature (T) at 2 m above ground (Table 4), about crop canopy height, were dependent on many factors, including spatial region of interest and topography."

RESPONSE: Revision adopted, as suggested. Some of your other suggestions also helped clarify pieces of this summary. The take-home message (last sentence) is that T datasets are pretty good except in mountains (another reference was added to demonstrate challenges in mountain regions).

Does the author define "adequate"?

RESPONSE: Not really (no specific criteria was applied even though determining "adequacy" of these datasets for hydrologic modelling was directly specified as an objective of the paper), but I deduce from the figure that seems to be used to support this statement that the Rocky Mountain region shows lower daily rank correlation and higher long-term bias, so that statement was added parenthetically: "…less adequate (lower daily rank correlation, higher long-term bias) in the Rocky Mountains."

This summary statement is helpful. Would recommend adding the number of studies ("consensus from these X studies") to this sentence.

RESPONSE: Clarified as follows (added underlined text): "The consensus from these nine studies (discussed in this section and cited in Table 4) suggests…". Two studies (Behnke, Tercek) did not have a hydrologic modeling component (and are not cited in Table 4), so did not inform this consensus. For further clarified, refined the Table 4 caption (with underlined text): "…literature on gridded dataset comparisons for hydrologic modelling".

The laundry list problem again - consider adding another summary statement to this section (at the top or bottom). One option would be to frame it from the perspective of a reader looking for the best dataset for their application (i.e., factors X, Y and Z have the biggest impact on dataset accuracy/are most important when choosing a data product).

RESPONSE: Added a concluding summary paragraph to this section: "Overall, the literature suggests the interaction of station density and basin characteristics, primarily topography, is of central importance and can drive performance. In regions with high station density (> 3 per 1000 km$^2$), G datasets or those corrected using G data (SG, RG, SRG, RSG) perform similarly. However, in areas with lower station density (< 1 per 1000 $km^2$) as well as in higher elevations and topographically complex regions, R datasets perform better. Unadjusted S datasets, without G or R correction, generally were least reliable. Other site and dataset considerations may also be important for specific hydrologic modelling applications and are discussed in the following sections."

Adequate for Q simulation at what river scale?

RESPONSE: Added the range of scales from these studied: "…and spatial scales ranging from 3,000 to 122,000 $km^2$".

"detecting" - unclear word choice. Possibly mean representing?

RESPONSE: Edited as suggested.

Could use another summary statement for this section.

RESPONSE: Added additional results from Blankenau et al. (2020) and Ang et al. (2022) to this section as well as a summary statement: "These results indicate that accuracy in climate data translated into accuracy in ET and SWE simulation and suggest that all gridded data be scrutinized, and possibly bias corrected, before use in ET and SWE modeling."

The Hydrologic Coherence Test sounds interesting! Consider including a brief example of how it would be used.

RESPONSE: Perhaps interesting, but perhaps somewhat simplistic. It's basically looking at independently calibrated model results from various gridded datasets, seeing which one(s) met a given model performance metric (any metric can be used… NSE, KGE, bias, etc.), and excluding the ones that didn't perform well enough. Basically, if a gridded dataset can't reproduce the streamflow (or other hydrologic outcome) results, then it should be excluded. I tried to use as few words as possible (added words are underlined) to explain this: "Laiti et al. (2018) demonstrated a Hydrologic Coherence Test (HyCoT), essentially a metric-independent method of comparing gridded datasets according to their performance in a hydrologic model, to exclude meteorological data less capable of reproducing a hydrologic outcome."

"similar climatic setting and hydrologic objectives" - similar to what? Similar to the reader's own research project? Or is this a recommendation to pay more attention to studies that have higher similarity within the datasets they consider?

RESPONSE: The former. Revised: "…with similar climatic setting and hydrologic objectives to the planned investigation."

This is a long and useful list of considerations. Consider adding "especially in areas of high topographic relief" after "derived from ground-based observations".

RESPONSE: Edited as suggested.

Consider adding vertical lines to Table A1 to visually separate columns. The word "Data" in "Network Common Data Form" runs into the descriptive text to its right.

RESPONSE: Agreed. I am certain this will be addressed in the final proof editing process.